# The Power of Random Features and
# the Limits of Distribution-Free Gradient Descent

**Ari Karchmer** [1]   **Eran Malach** [1]

## Abstract

We study the relationship between gradient-based optimization of parametric models (e.g., neural networks) and optimization of linear combinations of random features. Our main result shows that if a parametric model can be learned using mini-batch stochastic gradient descent (bSGD) without making assumptions about the data distribution, then with high probability, the target function can also be approximated using a polynomial-sized combination of random features. The size of this combination depends on the number of gradient steps and numerical precision used in the bSGD process. This finding reveals fundamental limitations of distribution-free learning in neural networks trained by gradient descent, highlighting why making assumptions about data distributions is often crucial in practice. Along the way, we also introduce a new theoretical framework called average probabilistic dimension complexity (adc), which extends the probabilistic dimension complexity developed by Kamath et al. (2020). We prove that adc has a polynomial relationship with statistical query dimension, and use this relationship to demonstrate an infinite separation between adc and standard dimension complexity.

## 1. Introduction

Theoretically, learning neural networks is computationally hard in the worst case. However, the practice of modern deep learning has demonstrated the remarkable power of gradient-based optimization of neural networks. Why does gradient descent over neural networks work so well in practice, despite theoretical pessimism? A full answer to this question is still missing in the science of deep learning.

One possible source of the computational hardness in theoretical studies is the goal of *distribution-free* learning. In distribution-free learning, the goal is to design algorithms that can learn a target function $f$ from a class $\mathcal{F}$ without making strong assumptions about the distribution of the input examples. Specifically, let $\rho$ be an arbitrary distribution over the input domain, and let $(x, f(x))$ be an example drawn from $\rho$. A distribution-free learning algorithm must succeed in approximating $f$ for any choice of $\rho$. This stands in contrast to distribution-specific learning, where the algorithm is designed with prior knowledge about the input distribution, often leading to more efficient and practical solutions. While distribution-free learning is a desirable goal, it is often computationally challenging, as it requires the algorithm to handle worst-case scenarios.

**This work, in a nutshell.** In this work, we discover a surprising relationship between distribution-free learning with gradient descent over differentiable parametric model classes and random feature representations. Informally speaking, we show the following implication. Let $\mathcal{F}$ be a class of target functions. Assume that a differentiable parametric model class can be successfully optimized in a distribution-free manner by mini-batch gradient descent to approximate all $f \in \mathcal{F}$. Let $\mu$ be a *target distribution* over $\mathcal{F}$. Then, with high probability over $f \sim \mu$, $f$ can be approximated by a linear combination of random features. Importantly, the number of random features necessary to include in this linear combination is upper bounded by a *polynomial* function of the number of optimization steps of gradient descent.

This result, as stated, sounds like a very strong positive result: could we now replace SGD on complex networks with optimization of simpler random features model? This is the positive interpretation of the result. However, we know that random features are limited in their expressive power. So, another (and we believe better) way to understand this result is as a statement on the limitation of distribution-free SGD. Indeed, our result, more formally described on the next page, reveals a fundamental limitation of distribution-free learning:

> If a function class is learnable in a distribution-free manner by gradient descent, then most func-

[1]Harvard University, Cambridge, Massachusetts, USA. Correspondence to: Ari Karchmer <akarchmer0@gmail.com>.

*Proceedings of the $42^{nd}$ International Conference on Machine Learning*, Vancouver, Canada. PMLR 267, 2025. Copyright 2025 by the author(s).

tions in the class must have a relatively simple random feature representation.

This limitation has a clear connection to the practice of deep learning, as we discuss next. Specifically, our result provides a theoretical foundation for the observation that distributional assumptions are crucial for the success of gradient-based optimization.

**The theoretical value of this result.** A common finding in the practice of Machine Learning is that making appropriate distributional assumptions often leads to dramatically better learning outcomes. For example, it has been shown both empirically and theoretically that, when examples are drawn from the uniform distribution, learning parity functions—Boolean functions that compute the parity (XOR) of a subset of input bits—can be extremely difficult for gradient descent on neural networks. However, the same task becomes *much* easier under biased product distributions over the input domain (see e.g., Malach & Shalev-Shwartz (2019)).

Parity functions are a cornerstone of computational learning theory due to the fact that, despite their simplicity, they exhibit exponential hardness under many learning frameworks (e.g., statistical queries and random features). Parity functions are also useful as informative special cases when studying difficult topics in deep learning theory. For example, parity functions have been recently used to study length complexity of chain-of-thought in autoregressive models (Malach, 2023), hidden progress in gradient descent (Barak et al., 2022), and pareto frontiers in width, initialization and computation of gradient descent (Edelman et al., 2023).

In contrast, our result helps *explain* the phenomenon, instead of just *demonstrating* it for the case of learning parity functions. We bring to light the fact that distribution-free learning with gradient descent essentially "collapses" to learning linear combinations of random features, in the average case. To ground this in the specific case of parity functions, we comment that random parities are well known to be computationally hard to learn using linear combinations of random features (even in the average case). Thus, our result provides a *consistent* explanation for why distribution-**free** learning of parities with gradient descent is hard, in terms of the well-known hardness of learning parities with random features. Again, note that random parities are *not* hard to learn with gradient descent in the distribution-**specific** case.

**Main theorem in more detail.** Let us now describe the main theorem contributed by this work in more detail. We consider a differentiable parametric hypothesis class $\mathcal{H}$, which has $p$ parameters $\mathbf{w} = (w_1 \cdots w_p) \in \mathbb{R}^p$ and a bounded output range. Let $f_\mathbf{w}$ be the function in $\mathcal{H}$ corresponding to parameters set to $\mathbf{w}$. Suppose that data is generated according to a *source distribution* $\mathcal{D}_{f,\rho}$, which samples

unlabeled data $x$ in a domain $X$, according to an unknown example distribution $\rho$, and labels the data with values in $\{\pm 1\}$, according to a target function $f : X \to \{\pm 1\}$.

Roughly, our main result is the following. Let $\mathcal{F}$ be a class of target functions and let $\Delta(X)$ be the set of distributions over domain $X$. Suppose it is possible to distribution-freely learn an accurate model $f_{\mathbf{w}^*} \in \mathcal{H}$ for some unknown source distribution $\mathcal{D}_{f,\rho} \in \{\mathcal{D}_{f,\rho} : f \in \mathcal{F}, \rho \in \Delta(X)\}$ (with respect to to squared loss, for any $f$ and $\rho$) by mini-batch SGD with parameters $T, c, p$. Then, given any prior distribution $\mu$ over $\mathcal{F}$, there exists a random feature distribution $\mathcal{E}$ such that:

- For random features $\phi_1 \cdots \phi_d \sim \mathcal{E}$, there exists a weight vector $\mathbf{v}$ such that

$$\sum_{i \leq d} \mathbf{v}_i \phi_i \approx f$$

  under 0/1 loss with respect to $\rho$, with probability $99/100$ (over $\mu$).

- It holds that $d \leq \text{poly}(Tp/c)$.

Here, $T$ is the number of clipped mini-batch gradient updates, $p$ is the number of parameters in the differentiable model, $c \in (0, 1]$ controls the granularity of the mini-batch gradient estimates. We note that the batch size must satisfy certain a lower bound roughly polylogarithmic in $Tp$ (for constant $c$).

At the end of the technical overview (section 3), we state the main theorem in full mathematical formality as Theorem 3.3.

**Techniques and further contributions.** To prove the main theorem, we use a series of transformations between different learning paradigms and complexity measures.

- First we use a transformation from bSGD algorithms (with a certain maximum precision—these can be thought of as noisy gradients) to statistical query (SQ) learning methods, which was proved by Abbe et al. (2021). This allows us to then relate $\text{poly}(Tp)$, a polynomial function of the product of parameters and bSGD steps, to the statistical query dimension of the class of target functions that label the source distribution. This is done using the characterization of Blum et al. (1994).

- Following this, we use ideas from communication complexity, particularly those related to discrepancy and the 2-party norm of a function, to relate statistical query dimension to the probability of being able to *weakly*

approximate a source distribution by sampling a *single* random feature (we call this the Random Feature lemma). This step employs new mathematical techniques that resemble the circuit learning algorithms of Karchmer (2024a;b).

- Next, we also introduce a new technique that uses *boosting* techniques, such as Adaboost (Freund & Schapire, 1997; Domingo et al., 2000), as constructive proofs of the fact that we can combine these weak approximators into a strong approximator: a linear combination of a relatively small number of random features.

Along the way, we introduce a new notion of dimension complexity called *average* probabilistic dimension complexity. This notion relaxes both the standard notion (Ben-David et al., 2002), and the probabilistic notion (Kamath et al., 2020). Average probabilistic dimension complexity precisely captures the quantity of interest in this paper: the number of features needed to approximate a given function class, with high probability, over some prior distribution over the function class. Thus, as a further contribution, we give an "infinite" separation between average probabilistic dimension complexity, and standard dimension complexity. Previously, only an exponential separation was known between the intermediate notion, probabilistic dimension complexity, and standard dimension complexity, from Kamath et al. (2020). Kamath et al. (2020) highlighted as an open problem to resolve whether or not an infinite separation exists between probabilistic and standard dimension complexity. Our relaxed notion of average probabilistic dimension complexity is sufficient for an affirmative resolution, but we also show that there may exist complexity-theoretic barriers to demonstrating that our relaxed notion is *necessary* for the separation.

## 2. Related Work

Many related works analyze the relationship between gradient descent on wide neural networks and linear learning with random features (LLRF). Much of this line of work is based on the following idea, which we now illustrate using a simple one-layer neural network:

$$R(x) \triangleq \sum_{i=1}^{d} v_i \sigma(\langle \mathbf{w}_i, x \rangle + b_i) \qquad (1)$$

The $d$ neurons are identified by their weights $\mathbf{w}_i$ and bias $b_i$, $\sigma$ is an activation function, and the top layer applies a linear combination by weights $u_i$.

When $d$ is really large and the weights are randomly initialized, it can be shown that applying gradient descent over

these parameters effectively changes the weight on the bottom layer, $\mathbf{w}_i, b_i$ by a tiny amount (Chizat et al., 2019). This means that when gradient descent for a small bounded number of steps is used to learn $R$, essentially, the bottom layers could have remained fixed after random initialization. Thus, in this case, learning the overparameterized network is effectively the same as learning the linear combination of some random features of the form $\sigma(\langle \mathbf{w}_i, x \rangle + b_i)$, for which known techniques suffice to prove optimal convergence. All in all, this allows Andoni et al. (2014); Daniely (2017); Du et al. (2018; 2019); Li & Liang (2018); Allen-Zhu et al. (2019) (among others) to derive provable guarantees for learning overparameterized neural networks by analyzing them as one would analyze linear learning with random features (LLRF).

It is worth mentioning that the Neural Tangent Kernel (NTK, Jacot et al. (2018)) is a bit more subtle. The NTK approach goes beyond considering the features of the top layer. Instead, NTK constructs a kernel matrix where each entry represents the dot product between the gradients of the network output with respect to the parameters, evaluated at two different data points. This kernel matrix effectively encodes the similarity between data points in the high-dimensional gradient feature space (the feature map representation is the gradient of the neural net with respect to the model parameters at a certain training point).

The main difference between this line of work and ours is that we show the *collapse* of the learning capabilities of distribution-free gradient descent, while the above line work uses a connection to LLRF to illustrate *how* gradient descent *can* learn.

## 3. Technical Overview and Theorem Statements

In this section we introduce and formally define the relevant technical settings, and finish with full statements of our main theorem and other results. After that, we provide a bird's eye view of our path to proving the main theorem.

Let $\mathcal{F}$ be a class of target concepts $f : X \rightarrow \{\pm 1\}$. Let $\mathcal{D}_{f,\rho} \in \mathcal{D}_{\mathcal{F}} = \{\mathcal{D}_f : f \in \mathcal{F}, \rho \in \Delta(X)\}$ be a source distribution which samples unlabeled data $x$ in a domain $X$, according to an example distribution $\rho$ over domain $X$, and labels the data according to some $f \in \mathcal{F}$. We consider generally speaking learning functions $f : X \rightarrow \{\pm 1\}$ over an input space $X$, with the objective of minimizing the population loss:

$$\mathcal{L}^{\mathcal{D}_{f,\rho}}(f) \triangleq \mathbb{E}_{(x,y)\sim\mathcal{D}_{f,\rho}}[\ell(f(x), y)]$$

with respect to a source distribution $\mathcal{D}_{f,\rho}$ over $X \times \{\pm 1\}$, where $\ell : \mathbb{R} \times \{\pm 1\} \rightarrow \mathbb{R}_{\geq 0}$ is a loss function.

In this work, we take the loss function to be either the

squared-loss: $\ell_{\text{sq}}(\hat{y}, y) = \frac{1}{2}(\hat{y}-y)^2$ or 0-1 loss: $\ell_{01}(\hat{y}, y) = \mathbf{1}[\hat{y} \neq y]$. We write $\mathcal{L}_{01}$ and $\mathcal{L}_{\text{sq}}$ to denote population loss.

### 3.1. Learning Paradigms

In this work, we consider learning of differentiable parametric hypothesis classes such as feed-forward neural networks. A $p$-parameter differentiable model $\mathbf{w} = (w_1, \cdots w_p)$ is a function $f_{\mathbf{w}} : X \rightarrow [-1, 1]$, and for every $x \in X$ there is a gradient $\nabla_{\mathbf{w}} f_{\mathbf{w}}(x)$ on almost every $\mathbf{w} \in \mathbb{R}^p$ (not including sets of measure 0). Note the bounded range of the mode. We say that $\mathcal{H}$ is a differentiable parametric hypothesis class with $p$ parameters if every $h \in \mathcal{H}$ is written as a differentiable model with $p$ parameters.

**Mini-batch stochastic gradient descent.** We consider learning of differentiable parametric hypothesis classes by mini-batch stochastic gradient descent (bSGD). Our setting is the same as Abbe et al. (2021) and we adopt some of their notation and their definitions.

For a differentiable $f_{\mathbf{w}}$, an initial distribution $\mathcal{W}$ over $\mathbb{R}^p$, a gradient precision $c \in (0, 1)$, a mini-batch size $b$, and a stepsize $\gamma$, the bSGD paradigm iteratively computes new parametric models at a timestep $t$ defined by $\mathbf{w}^{(0)} \sim \mathcal{W}$ and $\mathbf{w}^{(t+1)} \leftarrow \mathbf{w}^{(t)} - \gamma g_t$. For a mini batch $B_t \sim \rho^b$, the function $g_t$ is the $c$-approximate clipped gradient $\bar{\nabla}\ell_{B_t}(f_{\mathbf{w}^{(t)}}) \triangleq \frac{1}{b}\sum_{i=1}^{b}[\nabla \ell_{B_t}(f_{\mathbf{w}^{(t)}})]_1$ where $[\cdot]_1$ denotes the entry-wise clipping over values to the interval $[-1, 1]$. The clipped gradient in $[-1, 1]^p$ is a $c$-approximate rounding if each entry of the the clipped gradient is an integer multiple of $c$ and $||g_t - \bar{\nabla}\ell_{B_t}(f_{\mathbf{w}^{(t)}})||_{\infty} \leq 3c/4$.

We say that a learning algorithm $A$ is a $\text{bSGD}(T, c, b, p)$ method for a differentiable parametric hypothesis class with $p$ parameters if works by computed bSGD for $T$ updates on the $p$-parameter differentiable model, with $c$-approximate gradient clipping and batch size $b$, and beginning from initialization distribution $\mathcal{W}$. The algorithm $A$ ensures distribution-free $\epsilon$-accuracy if for any source distribution $\mathcal{D}_{f,\rho} \in \mathcal{D}_{\mathcal{F}}$, $f_{\mathbf{w}^{(t)}} \leftarrow A$ satisfies $\mathbb{E}\left[\sup \mathcal{L}^{\mathcal{D}_{f,\rho}}(f_{\mathbf{w}^{(t)}})\right] \leq \epsilon$. The expectation is taken over the random initialization of $f_{\mathbf{w}^{(0)}}$ and the selections of the mini-batches. The supremum is taken over all valid $c$-approximate gradient clippings.

We refer the reader to section 2 of Abbe et al. (2021) for discussion on some of the modelling/design choices taken in defining the bSGD model.

**Statistical query learning and SQ dimension.** We consider statistical query learning. A learning algorithm $A$ is said to be a $SQ(k, \tau)$ method if in $k$ iterations, it produces a statistical query $\phi_t : X \times \{\pm 1\} \rightarrow [-1, 1]$ at iteration $t$, which then gets a response $v_t$ from an oracle, and then after $k$ iterations outputs a candidate hypothesis $f : X \rightarrow \mathbb{R}$. Each successive statistical query $\phi_t$ is allowed to depend on

internal randomness and all prior statistical queries $\phi_r$ for $r < t$. The algorithm $A$ ensures distribution-free $\epsilon$-accuracy if for any source distribution $\mathcal{D}_{f,\rho} \in \mathcal{D}_{\mathcal{F}}$, $h \leftarrow A$ satisfies $\mathbb{E}\left[\sup \mathcal{L}^{\mathcal{D}_{f,\rho}}(h)\right] \leq \epsilon$, where the supremum is taken over all valid responses $v_t$ to each query $\phi_t$, which are those that satisfy $|\mathbb{E}_{\rho}[\phi_t(x, y)] - v_t| \leq \tau$. The expectation is taken over the internal randomness of $A$.

If statistical query responses have some maximum degree of precision $c$ (i.e., $v_t \in [-1, 1]$ is an integer multiple of $c \in (0, 1)$), then a statistical query $\phi : X \rightarrow [-1, 1]$ can be converted into a series of $t = \log(1/c)$ statistical queries $\Phi_1 \cdots \Phi_t : X \rightarrow \{\pm 1\}$ (note the Boolean range). This is done by asking individually for the expected value of each bit in the bit representation, and applying linearity to reconstruct the total expectation.

**Complexity of linear learning with random features.** Kamath et al. (2020) introduced a probabilistic variant of dimension complexity Ben-David et al. (2002) to in order to study the complexity of optimizing a linear combination over random features to fit a source distribution.

Let $\mathcal{H}$ be a hypothesis class consisting of functions of the form $h : X \rightarrow \mathbb{R}$, and $\ell$ be an implicit loss function (to be stated explicitly in context).

**Definition 3.1** (Probabilistic dimension complexity)**.** The quantity $\text{dc}_{\epsilon}(\mathcal{H})$ is the smallest positive integer $d$ such that there exists a distribution $\mathcal{E}$ over feature maps $\phi : X \rightarrow \mathbb{R}^d$, such that **for every** $\rho$ over $X$, **and every** $h \in \mathcal{H}$,

$$\mathbb{E}_{\phi \sim \mathcal{E}}\left[\inf_{\mathbf{w} \in \mathbb{R}^d} \mathcal{L}^{\mathcal{D}_{h,\rho}}(\langle \mathbf{w}, \phi \rangle)\right] \leq \epsilon \qquad (2)$$

We use the notation $\langle \mathbf{w}, \phi \rangle$ to denote the function defined $\langle \mathbf{w}, \phi \rangle(x) = \sum_i \mathbf{w_i}\phi(x)_i$.

Clearly, showing an upper bound on the probabilistic dimension complexity $d$ of a hypothesis class is *sufficient* to reduce learning source distribution to learning the optimal linear combination over the $d$-dimensional feature map implied by the upper bound.

**Complexity of linear learning with random features and a prior.** We introduce an average-case analogue of probabilistic dimension complexity. Let $\ell$ be an implicit loss function (to be stated explicitly in context).

**Definition 3.2** (Average probabilistic dimension complexity)**.** The quantity $\text{adc}_{\epsilon,\delta}(\mu)$, given a prior distribution $\mu$ over a hypothesis class $\mathcal{H}$, is the smallest positive integer $d$ such that there exists a distribution $\mathcal{E}$ over embeddings $\phi : X \rightarrow \mathbb{R}^d$, such that

$$\Pr_{h \sim \mu}\left[\forall \rho : \mathbb{E}_{\phi \sim \mathcal{E}}\left[\inf_{\mathbf{w} \in \mathbb{R}^d} \mathcal{L}^{\mathcal{D}_{h,\rho}}(h, \langle \mathbf{w}, \phi \rangle)\right] \leq \epsilon\right] \geq 1 - \delta$$

Clearly, showing an upper bound on the *average* probabilistic dimension complexity $d$ of a hypothesis class is sufficient to reduce learning source distribution to learning the linear combination over the $d$-dimensional feature map implied by the upper bound, which is optimal for a *large probability mass* of $\mathcal{H}$.

We discuss at length the motivation of average probabilistic dimension complexity in the context of the dimension complexity literature in the Appendix, Section B.

## 3.2. Main Theorem Statement

We can now state our main theorem, which demonstrates that, with respect to any prior distribution $\mu$, the learnability of *a large probability mass* of data generating source distributions that are efficiently accurately learnable by bSGD, could potentially be analyzed as efficiently accurately learnable by the LLRF method.

Let $T, b, p \in \mathbb{Z}$, $c \in (0, 1]$ be such that $bc^2 \geq \Omega(\log Tp/\delta)$. Let $\mathcal{H}$ be a differentiable parametric hypothesis class, where each model $f_\mathbf{w} : X \to [-1, 1]$ has $p$ parameters $\mathbf{w} = (w_1 \cdots w_p)$. Let $\mathcal{F}$ be a class of target concepts $f : X \to \{\pm 1\}$. Let $\mathcal{D}_{f,\rho} \in \mathcal{D}_\mathcal{F} = \{\mathcal{D}_{f,\rho} : f \in \mathcal{F}, \rho \in \Delta(X)\}$ be a source distribution which samples unlabeled data $x$ in a domain $X$, according to an example distribution $\rho$, and labels the data according to some $f \in \mathcal{F}$. Let $\mu$ be a prior distribution over $\mathcal{F}$.

Define $\mathrm{err}(A, \mathcal{D}_{f,\rho}) \triangleq \inf_\epsilon \left[ \mathbb{E} \left[ \sup_{h \leftarrow A} \mathcal{L}^{\mathcal{D}_{f,\rho}}(h) \right] \leq \epsilon \right]$. That is, the infimum over $\epsilon$ such that the algorithm $A$ ensures distribution-free $\epsilon$-accuracy on $\mathcal{D}_{f,\rho}$, where the expectation is over the random initialization of the differentiable model and the mini-batches. distribution-free $\epsilon$-accuracy on $\mathcal{D}_{f,\rho}$ is measured with respect to some loss function $\ell^\rho$. Let $\ell^\rho_{\mathrm{sq}}(\hat{y}, y) = \frac{1}{2}(\hat{y} - y)^2$. Let $\ell^\rho_{01}(\hat{y}, y) = \mathbf{1}[\hat{y} \neq y]$.

**Theorem 3.3** (Main theorem). *Suppose there exists a learning algorithm $A_{\mathrm{bSGD}}$ that is a $\mathrm{bSGD}(T, c, b, p)$ method, and $\mathrm{err}(A_{\mathrm{bSGD}}, \mathcal{D}_{f,\rho}) \leq 1/10$ for every source distribution $\mathcal{D}_{f,\rho} \in \mathcal{D}_\mathcal{F}$ with respect to $\ell^\rho_{\mathrm{sq}}$. Then, there exists positive integer $d \leq \mathrm{poly}(Tp/c^2)$ such that there exists a distribution $\mathcal{E}$ over embeddings $\phi : X \to \mathbb{R}^d$, such that for arbitrarily small constants $\epsilon, \delta > 0$:*

$$\Pr_{h \sim \mu} \left[ \forall \rho : \underset{\phi \sim \mathcal{E}}{\mathbb{E}} \left[ \inf_{\mathbf{w} \in \mathbb{R}^d} \mathcal{L}^{\mathcal{D}_{h,\rho}}_{01}(\langle \mathbf{w}, \phi \rangle) \right] \leq \epsilon \right] \geq 1 - \delta$$

*In other words, for any prior distribution $\mu$ over $\mathcal{F}$, $\mathrm{adc}_{\epsilon,\delta}(\mu) \leq \mathrm{poly}(Tp/c^2)$.*

**A Note on Gradient Precision.** It is worth mentioning that a result analogous to the above theorem, but for gradient descent with arbitrarily fine precision, is not possible. Indeed, our theorem would not hold if there were no restriction on the precision of the gradients. To see this, we can

borrow from the work of Abbe et al. (2021). They show in Theorem 1a of their paper that (distribution-free) PAC-learning algorithms can be simulated by (distribution-free) bSGD algorithms, when allowed fine enough gradient precision $c$. Specifically, when $c < 1/8b$. Using this theorem, it follows that for $b, c$ such that $c < 1/8b$, then bSGD can learn parities in the distribution-free case. From here, we can conclude that our transformation from bSGD to random features cannot hold for $b, c$ such that $c < 1/8b$, since the size of the implied random feature representation would violate SQ dimension lower bounds for parities.

## 3.3. Technical Tools

Now, we define some technical concepts, theorems, and quantities of interest that are useful in proving our results. Using those, we will then outline the path we will take to prove Theorem 3.3 via modular proof of the theorem.

**Relation Between SQ-Learning and bSGD.** Abbe et al. (2021) show that SQ-learning is as powerful as bSGD methods in the following sense.

**Theorem 3.4** (Thm 1c. of Abbe et al. (2021)). *Let $\ell(\hat{y}, y) = \frac{1}{2}(\hat{y} - y)^2$. Let $T, b, p \in \mathbb{Z}$, $c \in (0, 1]$ be such that $bc^2 \geq \Omega(\log Tp/\delta)$. For every learning algorithm $A_{\mathrm{bSGD}}$ that is a $\mathrm{bSGD}(T, c, b, p)$ method, there exists a learning algorithm $A_{\mathrm{SQ}}$ which is a $\mathrm{SQ}(Tp, c/8)$ method, such that for every source distribution $\mathcal{D}_{f,\rho}$,*

$$\mathrm{err}(A_{\mathrm{SQ}}, \mathcal{D}_{f,\rho}) \leq \mathrm{err}(A_{\mathrm{bSGD}}, \mathcal{D}_{f,\rho}) + \delta$$

In words, this theorem says that when setting batch-size and gradient precision appropriately, mini-batch SGD algorithms can be converted into SQ-learning algorithms that suffer only a small additive loss of accuracy, and the number of statistical queries is the product of the number of batch gradient updates and the size of the differentiable model.

**SQ-learning Characterized by SQ dimension.** The complexity of SQ-learning itself can also be captured by a simple combinatorial parameter called the SQ dimension (see Blum et al. (1994)).

Consider now a function $h : X \to \{\pm 1\}$ over a finite domain $X$. A hypothesis class $\mathcal{H}$ is a set of hypotheses $h : X \to \{\pm 1\}$.

**Definition 3.5** (Statistical query dimension). Let $\rho$ be a distribution over domain $X$. The statistical query dimension over $\rho$ of $\mathcal{H}$, denoted $\mathrm{sq}(\mathcal{H}, \rho)$, is the largest number $d$ such that there exists $d$ functions $f_1, \cdots f_d \in \mathcal{H}$ that satisfy, for all $i \neq j$:

$$\underset{x \sim \rho}{\mathbb{E}} [f_i(x) f_j(x)] \leq \frac{1}{d}$$

We define $\mathrm{sq}(\mathcal{H}) = \max_\rho \mathrm{sq}(\mathcal{H}, \rho)$.

The relationship between query complexity in the SQ-learning model and SQ dimension proved by Blum et al. (1994) is the following. We state their result using our notation and terminology.

**Theorem 3.6** (Blum et al. (1994)). *Let $d = \mathrm{sq}(\mathcal{H})$ and $\ell_{01}(\hat{y}, y) = \mathbf{1}[\hat{y} \neq y]$. If $A$ is a $SQ(k, \tau)$ method that is distribution-free $1/2 - \tau$-accurate, then $k > \frac{d\tau^2 - 1}{2}$.*

**Communication Complexity.** Our proofs use as tools many ideas from the theory of communication complexity. We will introduce the necessary notation, definitions and concepts in the Appendix, section D. We refer the reader to Kushilevitz & Nisan (1996) for more of the basics.

### 3.4. Separations Between Dimension Complexities

As a corollary of Theorem 4.1, we make some possible progress towards answering the open question left by Kamath et al. (2020). We recall their question was whether there exists a hypothesis class $\mathcal{H}$, which satisfies for 0/1 loss, and some functions $f, f' : \mathbb{N} \to \mathbb{N}$, both expressions $\mathrm{dc}(\mathcal{H}) \in O(f(n))$ and $\mathrm{dc}_\epsilon(\mathcal{H}) \in O(1/f'(\epsilon))$?

Our result, shows that the answer is yes, if we take $\mathrm{adc}_{\epsilon,\delta}^\ell$ instead of $\mathrm{dc}_\epsilon^\ell$.

**Corollary 3.7** (Infinite Separation between $adc_{\epsilon,\delta}^\ell$ and $dc^\ell$). *There exists a hypothesis class $\mathcal{H}$, with domain $\{\pm 1\}^n$ and range $\{\pm 1\}$, which satisfies for 0/1 loss, and **any** prior distribution $\mu$ over $\mathcal{H}$, and arbitrarily small constant $\delta > 0$:*

- $\mathrm{dc}(\mathcal{H}) \in 2^{\Omega(n^{\frac{1}{4}})}$

- $\mathrm{adc}_{\epsilon,\delta}(\mathcal{H}) \in O(1/\epsilon)$.

This separation follows immediately from Theorem 4.1 and a theorem of Sherstov (2008a), which is concerned with the family of Zarankiewicz matrices. We refer to the Appendix section A and B for the details and a definition of $\mathrm{dc}(\mathcal{H})$.

## 4. Modular Proof of Theorem 3.3

To prove Theorem 3.3, we use the relationships introduced in the previous section to reduce our goal to proving the following standalone theorem.

**Theorem 4.1.** *Let $\mathcal{F}$ be a function class, $\mu$ a distribution over $\mathcal{F}$, and let consider $\ell_{01}$ loss. We have:*

$$\mathrm{adc}_{\epsilon,\delta}(\mu) \leq O(\mathrm{sq}(\mathcal{F})^{24.01})$$

*where $\epsilon, \delta > 0$ are arbitrarily small constants.*

We will prove Theorem 4.1 in the next section. For now, we prove Theorem 3.3 assuming Theorem 4.1.

*Proof of Theorem 3.3.* First, observe that under the conditions of Theorem 3.3, Theorem 3.4 implies that for the assumed algorithm $A_{\mathrm{bSGD}}$, which is a $\mathrm{bSGD}(T, c, b, p)$ method, there exists a learning algorithm $A_{\mathrm{SQ}}$ which is a $SQ(Tp, c/8)$ method, such that for every source distribution $\mathcal{D}_{f,\rho}$,

$$\mathrm{err}(A_{\mathrm{SQ}}, \mathcal{D}_{f,\rho}) \leq \mathrm{err}(A_{\mathrm{bSGD}}, \mathcal{D}_{f,\rho}) + \delta.$$

Now, by Theorem 3.6, we know that since $A_{\mathrm{SQ}}$ is a $SQ(Tp, c/8)$ method that is distribution-free $\epsilon + \delta$-accurate, then $Tp > (dc^2/64 - 1)/2 > \Omega(dc^2)$, where $d = \mathrm{sq}(\mathcal{F})$. Rearranging, we conclude that $d < O(Tp/c^2)$.

We can now invoke Theorem 4.1 to conclude that $\mathrm{adc}_{\epsilon,\delta}(\mu) \leq \mathrm{poly}(Tp/c^2)$, where $\mathrm{adc}_{\epsilon,\delta}(\mu)$ is with respect to $\ell_{01}$. $\square$

**Remark.** The explicit conclusion of Theorem 3.3 is that with high probability over $f \sim \mu$, the source distribution can be $\epsilon$-approximated with respect to $\mathcal{L}_{01}^{\mathcal{D}_{f,\rho}}$ by a linear combination of $\mathrm{poly}(Tp/c)$ random features $\langle \mathbf{w}, \phi \rangle$. This implies that the corresponding **learned** parametric model $f_{\mathbf{w}^*}$ can also be $O(\epsilon)$- approximated with respect to $\mathcal{L}_{\mathrm{sq}}^{\mathcal{D}_{f,\rho}}$ by a linear combination of $\mathrm{poly}(Tp/c)$ random features. This follows from the fact that we are considering source distributions that have a deterministic labelling function and because the learned parametric model $f_{\mathbf{w}^*} : X \to [-1, 1]$ has a bounded range, which means that per-sample squared loss is at most 2.

## 5. Outline of Proof of Theorem 4.1

We present an outline of our proof of Theorem 4.1, which we defer in full to the appendix.

- (Step 1; Section A.1). We will apply a theorem of Sherstov (2008b), to conclude that statistical query dimension of a function class $\mathcal{F}$ controls the reciprocal of the correlation of $A$ (the sign matrix representation of $\mathcal{F}$) and 2-bit 2-party deterministic communication protocols. Here, correlation is measure with respect to a product distribution over the rows and columns of $A$. This induces a prior $\mu$ over the hypothesis class and a distribution $\rho$ over unlabeled inputs.

- (Step 2; Section A.2). We will use the results of step 1 and an analysis inspired by Karchmer (2024b) to show a *Random Feature lemma*. Our Random Feature lemma essentially says that, for every prior $\mu$, with high probability over target function $f \sim \mu$, then for every example distribution $\rho$, there exists a feature distribution $\mu_\rho^{feat}$ over *features*, such that $\mu_\rho^{feat}$ samples weak approximators for the target function $f$ (with respect

to $\rho$). The predictive accuracy of the weak approximator is $1/2 - \gamma$, where $\gamma$ is a polynomial function of $1/\mathrm{sq}(\mathcal{F})$.

- (Step 3; Section A.3). We will employ a new analysis technique, which uses *boosting* theorems such as Adaboost as constructive proofs of the fact that a relatively small linear combination of weakly predictive random features can approximate a given concept under the prior $\mu$.

## 6. Conclusion

Our results show that distribution-free gradient-based learning of parametric models collapses to optimization of linear combinations of random features, in the average case. This indicates limits on the capabilities of neural networks without distributional assumptions and explains why tasks such as parity learning remain hard under these conditions. We also introduce average probabilistic dimension complexity (adc), which admits an infinite separation from standard dimension complexity and clarifies the importance of distributional assumptions for gradient-based methods.

**Practical implications.** When designing learning algorithms, there is often a tension between making them work for all possible distributions (distribution-free) versus optimizing for expected scenarios (distribution-specific). Our result suggests that pursuing distribution-free guarantees may come at a substantial cost in terms of model expressiveness. This provides theoretical support for the common practice of incorporating domain knowledge and distributional assumptions into model design. What does this imply for the design of future learning algorithms? Our work suggests that embracing distributional assumptions may be key to unlocking the full potential of gradient-based optimization.

## Impact Statement

This paper presents work whose goal is to advance the field of Machine Learning. There are many potential societal consequences of our work, none which we feel must be specifically highlighted here.

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

# A. Proof of Theorem 4.1

In this section we will prove Theorem 4.1.

**Notation.** In all of the proof, we abuse notation and write $\mathrm{sq}(A)$ to denote the statistical query dimension of the function class represented by the sign matrix $A \in \{\pm 1\}^{|\mathcal{F}| \times |X|}$. This sign matrix has rows indexed my concepts $f \in \mathcal{F}$, and columns indexed by $x \in X$. For any function class $\mathcal{F}$, we will always write its sign matrix representation simply as $A$. We will use the notation $A(f; x)$ to represent the entry of $A$ at row $f$ and column $x$. We will also write $A(f; \cdot)$ to denote the entire row $f$, which can be viewed as a table of values of the concept $f \in \mathcal{F}$.

We refer to Appendix section D for more preliminaries on the communication complexity used in this section.

## A.1. Applying Sherstov's theorem

We make use of a theorem due to Sherstov (2008b). This theorem says that if the statistical query dimension of a sign matrix $A$ is polynomial, then the reciprocal of the discrepancy of $A$, minimized over product distributions over $\mathcal{F} \times X$, is also polynomial.

**Theorem A.1** (Sherstov (2008b) - Thm. 7.1)**.** *Let $A \in \{\pm 1\}^{|\mathcal{F}| \times |X|}$ be the sign matrix representation of a function class $\mathcal{F}$. We have that:*

$$\sqrt{\frac{1}{2}\mathrm{sq}(A)} \leq \frac{1}{\mathrm{disc}^{\times}(A)} \leq 8\mathrm{sq}(A)^2$$

**Lemma A.2.** *Let $A$ be a sign matrix. If $\mathrm{disc}^{\times}(A) \geq \gamma$, then there exists a 2-bit distributional communication protocol $\pi$ for $A$ over product distributions with correlation $\gamma$. In other words, for any product distribution $\zeta$,*

$$\left| \underset{(x,y)\sim\zeta}{\mathbb{E}} [\pi(x, y) A(x; y)] \right| \geq \gamma$$

Using Lemma A.2 (see proof in Appendix, section D) and Sherstov's theorem, we get the following combined lemma.

**Lemma A.3.** *Let $A \in \{\pm 1\}^{|\mathcal{F}| \times |X|}$ be the sign matrix representation of a function class $\mathcal{F}$. Let $\zeta \triangleq (\mu, \rho)$ be any product distribution over $\mathcal{F} \times X$, inducing a distribution over $A$. There exists a 2-bit distributional communication protocol $\pi$ for $A$ over $\zeta$ with correlation $1/8\mathrm{sq}(A)^2$. In other words, for any product distribution $\zeta$,*

$$\left| \underset{(f,x)\sim\zeta}{\mathbb{E}} [\pi(f, x) \cdot A(f; x)] \right| \geq \frac{1}{8\mathrm{sq}(A)^2} \tag{3}$$

*Proof.* By Theorem A.1, we know that $\mathrm{disc}^{\times}(A) \geq 1/8\mathrm{sq}(A)^2$. Then by Lemma A.2 expression (3) follows. □

## A.2. Random Feature lemma

We can use Lemma A.3 to prove our *Random Feature lemma*. Our Random Feature lemma essentially says that, for every prior $\mu$, with high probability over $f \sim \mu$, then for any example distribution $\rho$, there exists a feature distribution $\mu_\rho^{feat}$ which samples weak predictors for the target function $f$ (with respect to $\rho$).

We now state the Random Feature lemma.

**Lemma A.4** (Random Feature lemma)**.** *Let $A \in \{\pm 1\}^{|\mathcal{F}| \times |X|}$ be the sign matrix representation of a hypothesis class $\mathcal{F}$. Let $\mu$ be a distribution over $\mathcal{F}$, let $\ell$ be $0/1$ loss with respect to an example distribution $\rho$, and let $\delta > 0$. It holds that, with probability at least $1 - \delta$ over $f \sim \mu$, for all $\rho$, there exists $\mu_\rho^{feat}$ such that:*

$$\underset{f'\sim\mu_\rho^{feat}}{\mathrm{Pr}} \left[ \mathcal{L}_{01}^{\mathcal{D}_{f,\rho}}(f') \leq \frac{1}{2} - \Omega\left(\mathrm{sq}(A)^{-8}\right) \right] \geq \Omega\left(\mathrm{sq}(A)^{-8}\right)$$

*Proof.* We begin by considering the following procedure.

We will demonstrate that this procedure is a randomized predictor for the unknown target $f$ (which is sampled according to prior $\mu$) on an input point $z$ (which is sampled according to example distribution $\rho$). In other words, we will show that for

---

**Algorithm 1** `Predict`$(z)$

---

1: **input:** point $z$; example oracle access to function $f \sim \mu$, with respect to example distribution $\rho$.
2: Sample $g \sim \mu$.
3: Sample $\langle x, f(x) \rangle$ from example oracle.
4: **predict:** $g(z) \cdot g(x) \cdot f(x)$

---

the unknown target $f$, it holds that, for an $\epsilon$ to be defined later:

$$\Pr_{f \sim \mu, z \sim \rho, \texttt{Predict}} \left[ \texttt{Predict}(z) = f(z) \right] \geq \frac{1}{2} + \epsilon \tag{4}$$

After demonstrating (4), we will then use (4) to derive the conclusion of the present lemma.

**Towards (4).** Let $A \in \{\pm 1\}^{|\mathcal{F}| \times |X|}$ be the sign matrix representation of a function class $\mathcal{F}$. Let $\mu$ be a prior over $\mathcal{F}$, and let $\rho$ be an example distribution over $X$. We will show that,

$$\Pr_{f \sim \mu, z \sim \rho} \left[ \texttt{Predict}(z) = f(z) \right] \geq \frac{1}{2} + \frac{1}{O(\mathrm{sq}(A)^8)}$$

Consider the function $\texttt{Eval}(r, w)$, which takes as input random strings $r$ and $w$, and uses them to sample (a string representation of) $f \sim \mu$ and $x \sim \rho$, and then outputs $f(x)$. Recall that by Lemma A.2, there exists a 2-bit distributional communication protocol $\pi$ for the sign matrix $A$ over a product distribution $(\mu, \rho)$ with correlation $1/8\mathrm{sq}(A)^2$. In other words, for any product distribution $(\mu, \rho)$,

$$\left| \mathbb{E}_{f, x \sim (\mu, \rho)} \left[ \pi(f, x) \cdot A(f; x) \right] \right| \geq \frac{1}{8\mathrm{sq}(A)^2}$$

Therefore, by substitution,

$$\left| \mathbb{E}_{\substack{r, w \sim U \\ f, x \sim (\mu, \rho)}} \left[ \pi(f, x) \cdot \texttt{Eval}(r, w) \right] \right| \geq \frac{1}{8\mathrm{sq}(A)^2}$$

By Theorem D.6, for every function $F : \{0, 1\} \times \{0, 1\} \to \{\pm 1\}$,

$$\mathrm{Cor}(f, \Pi_c) = \max_{\pi \in \Pi_c} \left| \mathbb{E}_x \left[ f(x) \cdot \pi(x) \right] \right| \leq 2^c \cdot R_2(F)^{1/4} \tag{5}$$

for $x$ uniformly distributed over $\{0, 1\} \times \{0, 1\}$. Note, $\Pi_c$ is the class of $c$-bit communication protocols, which is defined in the Appendix, section D. Hence,

$$\frac{1}{O(\mathrm{sq}(A)^8)} \leq R_2(\texttt{Eval}) = \mathbb{E}_{\substack{r_1, r_2 \\ w_1, w_2 \sim U}} \left[ \prod_{\epsilon_1, \epsilon_2 \in \{1, 2\}} \texttt{Eval}(r_{\epsilon_1}, w_{\epsilon_2}) \right]$$

Manipulating the RHS we can see that:

$$\frac{1}{O(\mathrm{sq}(A)^8)} \leq \mathbb{E}_{\substack{f, g \sim \mu \\ z, x \sim \rho}} \left[ g(z) \cdot g(x) \cdot f(x) \cdot f(z) \right]$$

$$\leq \mathbb{E}_{\substack{\texttt{Predict} \\ z \sim \rho, f \sim \mu}} \left[ \texttt{Predict}(z) \cdot f(z) \right]$$

This implies that `Predict` is a randomized predictor for the unknown target $f$, which satisfies:

$$\Pr_{f \sim \mu, z \sim \rho, \texttt{Predict}} \left[ \texttt{Predict}(z) = f(z) \right] \geq \frac{1}{2} + \frac{1}{O(\mathrm{sq}(A)^8)} \tag{6}$$

**Using (4) to conclude lemma A.4.** We will combine (4) and a subtle "averaging" argument to conclude lemma A.4.

**Lemma A.5** ("Averaging argument," see lemma A.11 of Arora & Barak (2009))**.** *If a random variable $X \in [0, 1]$, and* $\mathbb{E}[X] = p$, *then for any* $c < 1$,

$$\Pr[X \le cp] \le \frac{1-p}{1-cp}$$

Let us consider Predict to be a *deterministic* function that maps $z$ and random bits $\mathbf{r}$ to a value in $\{\pm 1\}$. The random bits $\mathbf{r}$ encompass what is necessary to sample all random variables in the course of running Predict (e.g., $g \sim \mu$, and a sample from the example oracle). We can then conclude from (6) and lemma A.5 that there exists many "good" random strings:

$$\Pr_{f \sim \mu, \mathbf{r}} \left[ \Pr_{z \sim \rho} \left[ \text{Predict}^f(z; \mathbf{r}) = f(z) \right] \ge \frac{1}{2} + \frac{1}{O(\text{sq}(A)^8)} \right] \ge \frac{1}{O(\text{sq}(A)^8)}$$

Note that the deterministic version of Predict described so far uses random bits $\mathbf{r}$ to sample $x \sim \rho$ in line 1, but then would need oracle access to $f$ to obtain the value of $f(x)$. We need to remove this need, and can do so as follows. Observe that the prediction output of Predict is $g(z)g(x)f(x)$. Thus, the predictor predicts $g(z)$ and negates that prediction if and only if $g(x)f(x) = -1$. Now, $g(x)f(x)$ is a Bernoulli random variable, which can be considered independent of $z, g$ and $f$. The mean $p$ of this random variable depends on $\rho$. Hence, we can sample this random variable directly, instead of using oracle access to $f$. Thus, under this version of Predict, we include the random coins to sample the Bernoulli inside $\mathbf{r}$, and again apply lemma A.5 to now instead conclude:

$$\Pr_{f \sim \mu, \mathbf{r}} \left[ \Pr_{z \sim \rho} \left[ \text{Predict}_\rho(z; \mathbf{r}) = f(z) \right] \ge \frac{1}{2} + \frac{1}{O(\text{sq}(A)^8)} \right] \ge \frac{1}{O(\text{sq}(A)^8)} \tag{7}$$

Note that now Predict depends on $\rho$ but not the example oracle.

From here, we would like to show that:

$$\Pr_{f \sim \mu} \left[ \Pr_{\mathbf{r}} \left[ \Pr_{z \sim \rho} \left[ \text{Predict}_\rho(z; \mathbf{r}) = f(z) \right] \ge \frac{1}{2} + \epsilon \right] \ge \epsilon \right] \ge 1 - \delta \tag{8}$$

where $\delta$ is a small positive constant, and we use $\epsilon$ in place of $O(\text{sq}(A)^{-8})$ to streamline notation. Note that, (8) holds for any prior $\mu$.

To show this we assume, for the sake of contradiction, that the set of functions $f$ for which

$$\Pr_{\mathbf{r}} \left[ \Pr_{z \sim \rho} \left[ \text{Predict}_\rho(z; \mathbf{r}) = f(z) \right] \ge \frac{1}{2} + \epsilon \right] < \epsilon \tag{9}$$

has probability greater than $\delta$ under $\mu$. That is,

$$\Pr_{f \sim \mu} \left[ \Pr_{\mathbf{r}} \left[ \Pr_{z \sim \rho} \left[ \text{Predict}_\rho(z; \mathbf{r}) = f(z) \right] \ge \frac{1}{2} + \epsilon \right] < \epsilon \right] > \delta$$

Now, let $\mu'$ be a **new** distribution over functions $f$ defined as the conditional distribution of $\mu$ restricted to this "bad" set. Formally, for any set $S$ of functions,

$$\mu'(S) = \frac{\mu(S \cap \text{Bad})}{\mu(\text{Bad})}$$

where

$$\text{Bad} = \left\{ f \; \middle| \; \Pr_{\mathbf{r}} \left[ \Pr_{z \sim \rho} \left[ \text{Predict}_\rho(z; \mathbf{r}) = f(z) \right] \ge \frac{1}{2} + \epsilon \right] < \epsilon \right\}$$

Note that, since $\mu(\text{Bad}) > \delta$, $\mu'$ is a well-defined probability distribution.

We can now compute the joint probability under $\mu'$:

$$\Pr_{f \sim \mu', \mathbf{r}} \left[ \Pr_{z \sim \rho} \left[ \texttt{Predict}_\rho(z; \mathbf{r}) = f(z) \right] \geq \frac{1}{2} + \epsilon \right]$$

For each $f$ in the support of $\mu'$, we have:

$$\Pr_{\mathbf{r}} \left[ \Pr_{z \sim \rho} \left[ \texttt{Predict}_\rho(z; \mathbf{r}) = f(z) \right] \geq \frac{1}{2} + \epsilon \right] < \epsilon$$

Therefore,

$$\mathbb{E}_{f \sim \mu'} \left[ \Pr_{\mathbf{r}} \left[ \Pr_{z \sim \rho} \left[ \texttt{Predict}_\rho(z; \mathbf{r}) = f(z) \right] \geq \frac{1}{2} + \epsilon \right] \right] < \epsilon$$

We now apply this towards the contradiction. Inequality (7) states that for *any* distribution $\mu$ (including $\mu'$), we have:

$$\Pr_{f \sim \mu', \mathbf{r}} \left[ \Pr_{z \sim \rho} \left[ \texttt{Predict}_\rho(z; \mathbf{r}) = f(z) \right] \geq \frac{1}{2} + \epsilon \right] \geq \epsilon$$

This leads to a contradiction because under $\mu'$, the probability is less than $\epsilon$, whereas it must be at least $\epsilon$.

Therefore, the set Bad has small measure. Indeed, the contradiction implies that our assumption about the size of Bad must be false. Therefore, the measure of Bad under $\mu$ must be at most $\delta$:

$$\Pr_{f \sim \mu} \left[ \Pr_{\mathbf{r}} \left[ \Pr_{z \sim \rho} \left[ \texttt{Predict}_\rho(z; \mathbf{r}) = f(z) \right] \geq \frac{1}{2} + \epsilon \right] < \epsilon \right] \leq \delta$$

Equivalently, the probability that $f$ is such that

$$\Pr_{\mathbf{r}} \left[ \Pr_{z \sim \rho} \left[ \texttt{Predict}_\rho(z; \mathbf{r}) = f(z) \right] \geq \frac{1}{2} + \epsilon \right] \geq \epsilon$$

is at least $1 - \delta$:

$$\Pr_{f \sim \mu} \left[ \Pr_{\mathbf{r}} \left[ \Pr_{z \sim \rho} \left[ \texttt{Predict}_\rho(z; \mathbf{r}) = f(z) \right] \geq \frac{1}{2} + \epsilon \right] \geq \epsilon \right] \geq 1 - \delta$$

Now, sampling a string of random bits $\mathbf{r}$ and then hard-coding it into $\texttt{Predict}_\rho$ induces a distribution over functions. Note that, this distribution is potentially different for each example distribution $\rho$, because $\texttt{Predict}$ samples $\rho$. Therefore, we can take this distribution over functions as $\mu_\rho^{feat}$, and conclude that,

$$\Pr_{f \sim \mu} \left[ \forall \rho \, \exists \, \mu_\rho^{feat} \; : \; \Pr_{f' \sim \mu_\rho^{feat}} \left[ \mathcal{L}_{01}^{\mathcal{D}_{f,\rho}}(f') \leq \frac{1}{2} - \Omega \left( \mathrm{sq}(A)^{-8} \right) \right] \geq \Omega \left( \mathrm{sq}(A)^{-8} \right) \right] \geq 1 - \delta$$

$\square$

## A.3. Proof of Theorem 4.1 via Constructive Boosting

Now that we have the Random Feature lemma, we prove Theorem 4.1. In order to do so, we will use equivalence of weak-to-strong learning in the PAC-setting.

**Theorem A.6** (Adaboost—Freund & Schapire (1997) (see also Karbasi & Larsen (2024))). *Let $\mathcal{O}$ be an oracle that accepts an example distribution $\rho'$ over $X$ and returns a $\gamma$-weak learner $f^{wk}$ with respect to 0-1 loss. There exists an algorithm* BOOST *that, for any example distribution $\rho$, makes $\tilde{\Theta}(\gamma^{-2})$ queries to $\mathcal{O}$ and outputs $g : X \to \{\pm 1\}$ such that, for any $h \in \mathcal{H}$, and any $\epsilon, \delta > 0$,*

$$\Pr_{g \leftarrow \texttt{BOOST}^{\mathcal{O}}} \left[ \mathcal{L}^{\mathcal{D}_{h,\rho}}(g) \leq \epsilon \right] \geq 1 - \delta$$

*Here, $\tilde{\Theta}$ hides logarithmic factors in $1/\delta, 1/\epsilon, 1/\gamma$, and the VC dimension of the class that $\mathcal{O}$ outputs weak learners from. Moreover, the final hypothesis $g$ found by BOOST is of the form*

$$g(x) \triangleq \text{sign}\left(\sum_i^{\tilde{\Theta}(\gamma^{-2})} w_i f_i^{wk}(x)\right) \quad : \quad w \in \mathbb{R}, f_i^{wk} \leftarrow \mathcal{O}$$

*Proof of Thm. 4.1.* To prove the statement, we need to show that, given a prior $\mu$ over a function class $\mathcal{F}$, the smallest positive integer $d$ such that there exists a distribution $\mathcal{E}$ over embeddings $\phi : X \to \mathbb{R}^d$, such that

$$\Pr_{f \sim \mu}\left[\forall \rho : \mathop{\mathbb{E}}_{\phi \sim \mathcal{E}}\left[\inf_{\mathbf{w} \in \mathbb{R}^d} \mathcal{L}^{\mathcal{D}_{f,\rho}}(\text{sign}(\langle \mathbf{w}, \phi \rangle))\right] \leq \epsilon\right] \geq 1 - \delta$$

is at most $O(\epsilon^{-1}\text{sq}(\mathcal{F})^{16+\hat{\epsilon}})$ for arbitrarily small $\hat{\epsilon} > 0$. Specifically, we will show this when $\mathcal{E}$ is defined as the probabilistic process that samples embedding $\phi$ by sampling $f_1', \cdots f_d' \sim \mu_{\rho_1}^{feat}, \cdots \mu_{\rho_d}^{feat}$, for some $d \leq O(\text{sq}(\mathcal{F})^{24.01})$ and outputting their direct product $(f_1', \cdots f_d')$. Observe that the underlying example distribution may be different for each $f_i'$. We will illuminate how each $\rho_i$ is chosen.

To prove this, first let us apply the Random Feature lemma to the conditions of the present theorem, to conclude that,

$$\Pr_{f \sim \mu}\left[\forall \rho \, \exists \, \mu_\rho^{feat} \, : \, \Pr_{f' \sim \mu_\rho^{feat}}\left[\mathcal{L}_{01}^{\mathcal{D}_{f,\rho}}(f') \leq \frac{1}{2} - \Omega\left(\text{sq}(\mathcal{F})^{-8}\right)\right] \geq \Omega\left(\text{sq}(\mathcal{F})^{-8}\right)\right] \geq 1 - \delta$$

Establishing this essentially gives a weak learning algorithm for each example distribution $\rho$. To see this, observe that the above equation says that, with probability $1 - \delta$ over $f \sim \mu$, for every $\rho$, $f' \sim \mu_\rho^{feat}$ weakly approximates $f$ over $\rho$ with probability $\Omega(\text{sq}(\mathcal{F})^8)$. The weak approximator has a population loss $\frac{1}{2} - \gamma$ for $\gamma \geq \Omega(\text{sq}(\mathcal{F})^{-8})$. We call it a $\gamma$-weak approximator. Also, we call $f'$ that satisfy this event "good."

Under this weak approximator interpretation of the Random Feature lemma, it becomes clear that we should sample $f_1', \cdots f_d' \sim \mu_{\rho_1}^{feat}, \cdots \mu_{\rho_d}^{feat}$ for successive $\rho_i$ in accordance with a standard weak-to-strong accuracy boosting algorithm such as Adaboost.

If we do this, then by Theorem A.6, we can constructively derive, for any $\rho$, a linear combination $\mathbf{w}$ such that,

$$\mathcal{L}_{01}^{\mathcal{D}_{f,\rho}}(g) \leq \epsilon/2 \quad : \quad g(x) \triangleq \text{sign}\left(\langle \mathbf{w}, \phi \rangle\right) \tag{10}$$

given enough samples from $\mu_{\rho_1}^{feat}, \cdots \mu_{\rho_d}^{feat}$ for the appropriate $\rho_i$. Note, also from Theorem A.6, $\mathbf{w} \in \mathbb{R}^{\tilde{\Theta}(\gamma^{-2})}$. Thus, we will conclude the desired statement by upper bounding the number of samples we need from successive $\mu_{\rho_1}^{feat}$ (i.e., the quantity $d$) as follows.

Let $D$ denote random variable of the number of features we need to sample until we can find a linear combination of features $g$ to satisfy eq. (10). We compute the expected value of $D$. We can analyze by "rounds." At round 0, there is a pool of $Z = \tilde{O}(\gamma^{-2})$ example distributions which require a $\gamma$-weak approximator. Now, at each round $r$, suppose a single random feature $f'$ is sampled from $\mu_{\rho_r}^{feat}$ as a potential $\gamma$-weak approximator for $\rho_r$. We progress to the next round if a $\gamma$-weak approximator is sampled for $\rho_r$, and remove $\rho_r$ from the pool of distributions which needs an approximator. Hence, round $r$ indicates that $r$ $\gamma$-weak approximators have been found, and $Z - r$ remain in the pool. We terminate if we reach round $Z$.

At each round,

$$\Pr_{f' \sim \mu_\rho^{feat}}\left[\mathcal{L}_{01}^{\mathcal{D}_{f,\rho}}(f') \leq \frac{1}{2} - \Omega\left(\text{sq}(\mathcal{F})^{-8}\right)\right] \geq \Omega\left(\text{sq}(\mathcal{F})^{-8}\right)$$

Hence the expected number of samples from $\mu_\rho^{feat}$ needed to satisfy the event that at least one of the samples is "good" is $O(\text{sq}(\mathcal{F})^8)$. Then, since $Z = \tilde{O}(\gamma^{-2})$, the expected number of samples to get a "good" feature for each of the $Z$ example distribution is $O(\text{sq}(\mathcal{F})^8) \cdot O(\text{sq}(\mathcal{F})^{16}) = O(\text{sq}(\mathcal{F})^{24.01})$.

We thus conclude,

$$\Pr_{f \sim \mu}\left[\mathbb{E}\left[D\right] \leq O(\text{sq}(\mathcal{F})^{24.01})\right] \geq 1 - \delta$$

By the Markov inequality,

$$\Pr_{f \sim \mu} \left[ \Pr_{D} \left[ D \geq \frac{2}{\epsilon} \cdot \Omega(\mathrm{sq}(\mathcal{F})^{24.01}) \right] \leq \epsilon/2 \right] \geq 1 - \delta$$

Combining this with (10), we have:

$$\Pr_{f \sim \mu} \left[ \forall \rho : \; \mathbb{E}_{\phi \sim \mathcal{E}} \left[ \inf_{\mathbf{w} \in \mathbb{R}^d} \mathcal{L}^{\mathcal{D}_{f,\rho}}(\mathrm{sign}(\langle \mathbf{w}, \phi \rangle)) \right] \leq \epsilon \right] \geq 1 - \delta$$

for $d \leq O(\frac{1}{\epsilon} \mathrm{sq}(\mathcal{F})^{24.01})$. $\qquad\qquad\square$

## B. Average Probabilistic Dimension Complexity

### B.1. Background

Towards understanding the limits or the power of learning linear combinations over features, one of the fundamental quantities of interest is the *dimension complexity* of a hypothesis class (see e.g. Ben-David et al. (2002)). Roughly, the dimension complexity of a hypothesis class $\mathcal{H}$ full of functions $h : X \to \mathbb{R}$ is the least positive $d \in \mathbb{Z}$ such that there exists a feature map $\phi : X \to \mathbb{R}^d$, such that for every $h \in \mathcal{H}$ there exists a linear combination $\mathbf{w} \in \mathbb{R}^d$ which satisfies $h(x) = \langle \mathbf{w}, \phi(x) \rangle$ for all $x \in X$.

More formally:

**Definition B.1** (Standard dimension complexity). The quantity $\mathrm{dc}(\mathcal{H})$ is the smallest positive integer $d$ such that there exists a feature map $\phi : X \to \mathbb{R}^d$, such that **for every** $\rho$ over $X$, **and every** $h \in \mathcal{H}$,

$$\inf_{\mathbf{w} \in \mathbb{R}^d} \mathcal{L}^{\mathcal{D}_{h,\rho}}(\langle \mathbf{w}, \phi \rangle) \tag{11}$$

Showing an upper bound on the dimension complexity $d$ of a hypothesis class is thus *sufficient* to reduce a Machine Learning problem to linear learning over the $d$-dimensional feature map implied by the upper bound. The smaller is $d$, the more efficient learning can be (in terms of sample complexity, for example).

However, this standard notion of dimension complexity is in some sense "overkill" for this reduction. It was pointed out by Kamath et al. (2020) that the standard dimension complexity is not *necessary* for Machine Learning, since for effective Machine Learning, one only needs to design a feature map that allows for $\epsilon$-approximation by a linear combination of the features, for each function in the hypothesis class. This also means that not only are upper bounds on dimension complexity overkill, but lower bounds do not necessarily rule out efficient learning.

### B.2. Probabilistic dimension complexity

In light of this, Kamath et al. (2020) introduced probabilistic variants of dimension complexity to get closer to the notion of dimension complexity that is necessary and sufficient for this reduction. Mainly, their definition subs out exact representation by a linear combination in favor of $\epsilon$-approximation. The following is their definition:

**Definition B.2** (Probabilistic dimension complexity). The quantity $\mathrm{dc}_\epsilon(\mathcal{H})$ is the smallest positive integer $d$ such that there exists a distribution $\mathcal{E}$ over feature maps $\phi : X \to \mathbb{R}^d$, such that **for every** $\rho$ over $X$, **and every** $h \in \mathcal{H}$,

$$\mathbb{E}_{\phi \sim \mathcal{E}} \left[ \inf_{\mathbf{w} \in \mathbb{R}^d} \mathcal{L}^{\mathcal{D}_{h,\rho}}(\langle \mathbf{w}, \phi \rangle) \right] \leq \epsilon \tag{12}$$

Only requiring $\epsilon$-approximation also means that the definition can allow for a distribution $\mathcal{E}$ over feature maps (for free, in a sense). This further opens up the possibility of using probabilistic dimension complexity to study the LLRF method.

The immediate question concerning probabilistic dimension complexity is whether it can be significantly smaller than standard dimension complexity. This is an important question because a yes answer would present a hypothesis class for which a lower bound on its dimension complexity would not necessarily rule out the effectiveness of the LLRF method. Indeed, Kamath et al. (2020) demonstrate this very possibility by giving an exponential *separation* between the probabilistic and standard dimension complexity.

**Theorem B.3** (Thm 6. of Kamath et al. (2020)). *There exists a hypothesis class $\mathcal{H}$, with domain $\{\pm 1\}^n$ and range $\{\pm 1\}$, which satisfies for 0/1 loss:*

- $\mathrm{dc}(\mathcal{H}) \in 2^{\Omega(n^{\frac{1}{4}})}$

- $\mathrm{dc}_\epsilon(\mathcal{H}) \in O(n^4/\epsilon)$.

It remains unknown whether there exists an "infinite" separation between standard dimension complexity and probabilistic dimension complexity. In fact, Kamath et al. (2020) explicitly leave it as an open question: does there exists a hypothesis class $\mathcal{H}$, which satisfies for 0/1 loss, and some functions $f, f' : \mathbb{N} \to \mathbb{N}$, $\mathrm{dc}(\mathcal{H}) \in O(f(n))$ and $\mathrm{dc}_\epsilon(\mathcal{H}) \in O(1/f'(\epsilon))$?

### B.3. Average Probabilistic Dimension Complexity

The definition of probabilistic dimension complexity maintains universal quantification over possible hypotheses $h \in \mathcal{H}$ (this quantifier was unchanged from standard dimension complexity). In other words, there still needs to exist one distribution over feature maps such that *for every $h \in \mathcal{H}$*, (12) holds.

As we alluded to previously, in this work, we point out that while universal quantification of $\mathcal{H}$ is desirable, this also makes it easier to prove a lower bound by constructing a pathological Machine Learning problem, when at the same time it is possible that all other instances of the Machine Learning problem have significantly lower complexity. This would make the lower bound hard to apply in practical situations—rarely would hypothesis functions be chosen "adversarially," (wherein universal guarantees would be the go-to).

In summary, while (Kamath et al. (2020) argued that) standard dimension complexity is overkill for a reduction to the LLRF method, we now point out that probabilistic dimension complexity might still be overkill for a reduction to the LLRF method in practical situations.

A more practical setting might be the average-case or "Bayesian view," where the learning problem is not necessarily chosen by an adversary (universal quantification), but instead by a randomized process known to the learner. The randomized process is the *prior* for the learner. Hence, we introduce a further relaxation of dimension complexity, where the new goal is probabilistic guarantees over **both** the accuracy, **and** the hypothesis class.[1] We define dimension complexity in this setting as follows:

**Definition B.4** (Average probabilistic dimension complexity). The quantity $\mathrm{adc}_{\epsilon,\delta}(\mu)$, given a prior distribution $\mu$ over a hypothesis class $\mathcal{H}$, is the smallest positive integer $d$ such that there exists a distribution $\mathcal{E}$ over embeddings $\phi : X \to \mathbb{R}^d$, such that

$$\Pr_{h \sim \mu} \left[ \forall \rho : \mathbb{E}_{\phi \sim \mathcal{E}} \left[ \inf_{\mathbf{w} \in \mathbb{R}^d} \mathcal{L}^{\mathcal{D}_{h,\rho}}(\langle \mathbf{w}, \phi \rangle) \right] \le \epsilon \right] \ge 1 - \delta$$

After defining average probabilistic dimension complexity, we would hope to find hypothesis class that has very low average probabilistic dimension complexity—even lower than its probabilistic dimension complexity, and hopefully much lower than standard dimension complexity. This would formally motivate our definition, and it would shed light on when the LLRF method might still be effective on **most** instances of a Machine Learning problem, or when the learner has an accurate prior.

## C. Results on Average Probabilistic DC vs. Standard DC

Consider hypothesis functions $h : X \to \{\pm 1\}$ over a finite domain $X$. A hypothesis class $\mathcal{H}$ is a set of concepts $h : X \to \{\pm 1\}$. As mentioned previously, one of our main theorems proves that average probabilistic dimension complexity is polynomially related to statistical query dimension. Thus we recall the definition of statistical query dimension.

**Definition C.1** (Statistical query dimension). Let $\rho$ be a distribution over domain $X$. The statistical query dimension over $\rho$ of $\mathcal{H}$, denoted $\mathrm{sq}(\mathcal{H}, \rho)$, is the largest number $d$ such that there exists $d$ functions $f_1, \cdots f_d \in \mathcal{H}$ that satisfy, for all $i \ne j$:

$$\mathbb{E}_{x \sim \rho} [f_i(x) f_j(x)] \le \frac{1}{d}$$

We define $\mathrm{sq}(\mathcal{H}) = \max_\rho \mathrm{sq}(\mathcal{H}, \rho)$.

---

[1]We can also view the average probabilistic dimension complexity as probabilistic dimension complexity of a **large probability mass** of hypotheses in the class $\mathcal{H}$ (with respect to the prior $\mu$).

We recall our main theorem from the body of the paper.

**Theorem C.2** (Restated Theorem 4.1). *Let $\mathcal{H}$ be a hypothesis class, $\mu$ a distribution over $\mathcal{H}$, and let $\ell$ be $0 - 1$ loss. We have:*

$$\mathrm{adc}_{\epsilon,\delta}(\mu) \leq O(\mathrm{sq}(\mathcal{H})^{24.01})$$

*where $\epsilon, \delta > 0$ are arbitrarily small constants.*

**Infinite separation between $adc_{\epsilon,\delta}^{\ell}$ and $dc^{\ell}$.** As an application of Theorem 4.1, we make some possible progress towards answering the open question left by Kamath et al. (2020). We recall there question was whether there exists a hypothesis class $\mathcal{H}$, which satisfies for 0/1 loss, and some functions $f, f' : \mathbb{N} \to \mathbb{N}$, $\mathrm{dc}(\mathcal{H}) \in O(f(n))$ and $\mathrm{dc}_\epsilon(\mathcal{H}) \in O(1/f'(\epsilon))$?

Our result, shows that the answer is yes, if we take $adc_{\epsilon,\delta}^{\ell}$ instead of $dc_\epsilon^{\ell}$.

**Corollary C.3** (Infinite Separation between $adc_{\epsilon,\delta}^{\ell}$ and $dc^{\ell}$). *There exists a hypothesis class $\mathcal{H}$, with domain $\{\pm 1\}^n$ and range $\{\pm 1\}$, which satisfies for 0/1 loss and **any** prior distribution $\mu$ over $\mathcal{H}$, and arbitrarily small constant $\delta > 0$:*

- $\mathrm{dc}(\mathcal{H}) \in 2^{\Omega(n^{\frac{1}{4}})}$

- $\mathrm{adc}_{\epsilon,\delta}(\mathcal{H}) \in O(1/\epsilon)$.

This separation follows immediately from Theorem 4.1 and a theorem of Sherstov (2008a), which is concerned with the following family of matrices.

**Definition C.4** (Zarankiewicz Matrices). Let $\mathcal{Z}(N, c)$ denote the class of $N \times N$ sign matrices that never have any $c \times c$ sub-matrix with all of its entries set to 1.

**Theorem C.5.** *Let $\epsilon > 0$ be an arbitrary constant. We have,*

$$\forall A \in \mathcal{Z}(N, 2\lceil \epsilon \rceil) \ : \ \mathrm{sq}(A) \in O(1)$$
$$\exists A \in \mathcal{Z}(N, 2\lceil \epsilon \rceil) \ : \ \mathrm{dc}(A) \in \Omega(N^{1-\epsilon})$$

## C.1. Towards Infinite Separation Between Probabilistic DC and Standard DC

In the previous section, we used Theorem 4.1, to prove an infinite separation between average probabilistic dimension complexity and dimension complexity. On the other hand, for probabilistic dimension complexity, Kamath et al. (2020) gave "just" an exponential separation ($O(n)$ vs. $2^{\Omega(n)}$). Thus, in this section, we consider whether or not it is possible to improve the separation of Kamath et al. (2020) to infinite. This was posed explicitly as an open problem by Kamath et al. (2020).

Said differently, we would like to understand if the relaxation from probabilistic dimension complexity to *average* probabilistic dimension complexity is *necessary* (and not merely *sufficient*) to prove an infinite separation with respect to dimension complexity.

As a first try towards proving that the relaxation is indeed necessary, we might try to again use Zarankiewicz matrices to show that there exists $A \in \mathcal{Z}(N, c)$ such that $\mathrm{dc}_\epsilon(A) \in \Omega(f(N))$ for a super-constant function $f$.

Unfortunately, if we go deeper in understanding Zarankiewicz matrices, we can see that this approach is not viable. Towards Theorem C.5, Sherstov uses a result of Ben-David et al. (2002) to obtain the dimension complexity lower bound on a matrix in the Zarankiewicz family. However, as Sherstov (2008a) notes, Ben-David et al. (2002) actually something stronger. They show that, for a fixed $c \in \mathbb{Z}$, *all but vanishing fraction* of $\mathcal{Z}(N, c)$ have dimension complexity $\Omega(N^{1-2/c})$. This implies that, given $\epsilon = 0$,

$$\Pr_{A \sim \mathrm{Unif}(\mathcal{Z}(N,c))} [\mathrm{dc}_\epsilon(A) \in \Omega(f(N))] \geq 0.99 \tag{13}$$

for $f(N) = N^{1-\frac{2}{c}}$.

Therefore, a *random* Zarankiewicz matrix has large dimension complexity. As a result, trying to prove (13) for $\epsilon > 0$ is actually impossible, even for $f \in o(N^{1-2/c})$. To see this, observe that if we did show that, then this would essentially be a proof of $\mathrm{adc}_{\epsilon,\delta}(\mathrm{Unif}(\mathcal{Z}(N,c))) \in \Omega(f(N))$. By Theorem 4.1, we now understand this statement to be false, for super-constant $f$!

## C.2. Barrier to Separation of Average Probabilistic DC and Probabilistic DC

Beyond the case of Zarankiewicz matrices, in general, it seems it will be difficult to prove that there exists $\mathcal{H}, \mu$ such that

$$\text{adc}_{\epsilon,\delta}(\mu)^{\omega(1)} < \text{dc}_\epsilon(\mathcal{H}) \tag{14}$$

In this section, we formalize this by demonstrating a barrier to proving (14), if we also restrict the separation to occur on a bounded $L_1$-norm version of average probabilistic dimension complexity.

**Definition C.6** (Bounded norm $\text{adc}_{\epsilon,\delta}(\mu)$). The quantity $\text{adc}^b_{\epsilon,\delta}(\mu)$, given a prior $\mu$ over a hypothesis class $\mathcal{H}$, is the smallest positive integer $d$ such that there exists a distribution $\mathcal{E}$ over embeddings $\phi : X \to \mathbb{R}^d$, such that

$$\Pr_{h \sim \mu}\left[ \forall \rho : \mathbb{E}_{\phi \sim \mathcal{E}}\left[ \inf_{\substack{\mathbf{w} \in \mathbb{R}^d \\ ||\mathbf{w}||_1 \leq b}} \mathcal{L}^{\mathcal{D}_{h,\rho}}(\langle \mathbf{w}, \phi \rangle) \right] \leq \epsilon \right] \geq 1 - \delta$$

Here, $||\mathbf{w}||_1 \triangleq \sum_i |\mathbf{w}_i|$.

We note that both Theorem 3.3 and the separation (Corollary C.3) actually apply to $\text{adc}^b_{\epsilon,\delta}(\mu)$, for $b \leq \text{poly}(\text{sq}(\mathcal{H}))$.

For now, we will prove a *complexity-theoretic* barrier, based on the long-time difficulty of proving super polynomial threshold circuit size lower bounds.

**Theorem C.7** (Complexity-theoretic barrier). *Let $b \leq 2^{n^{1-\hat{\epsilon}}}$ for some constant $0 < \hat{\epsilon} < 1$. Suppose that*

$$\text{adc}^b_{\epsilon,\delta}(\mu)^{\omega(1)} < \text{dc}_\epsilon(\mathcal{H})$$

*holds. Then, depth-2 threshold circuits computing* Eval$(\mathcal{H}) : (\hat{h}, \hat{x}) \to h(x)$ *require superpolynomial size (where $h, x$ are binary encodings of $h \in \mathcal{H}$ and $x \in X$).*

To do so, we use a theorem due to Alman & Williams (2017) which relates circuits size of depth-2 threshold circuits of evaluating a hypothesis class, and the probabilistic dimension complexity of the class.

**Lemma C.8** (Alman & Williams (2017)). *If* Eval$(\mathcal{H}) : (\hat{h}, \hat{x}) \to h(x)$ *(where $h, x$ are binary encodings of $h \in \mathcal{H}$ and $x \in X$) is computable by a depth-2 threshold circuit of size $s$, then:*

$$\text{dc}_\epsilon(\mathcal{H}) \leq O\left( \frac{s^2 \log^2(|\mathcal{H}| \cdot |X||)}{\epsilon} \right)$$

*Proof of Theorem C.7.* If it is true that, for $b \leq 2^{n^{1-\hat{\epsilon}}}$, for every $\mathcal{H}$ and $\mu$ over $\mathcal{H}$, there exists some constant $\gamma > 0$ such that,

$$\text{vc}(\mathcal{H})^\gamma \leq \text{adc}^b_{\epsilon,\delta}(\mu)$$

then, if we prove that there exists $\mathcal{H}, \mu$ such that

$$\text{adc}^b_{\epsilon,\delta}(\mu)^{\omega(1)} < \text{dc}_\epsilon(\mathcal{H})$$

then we deduce that,

$$\text{vc}(\mathcal{H})^{\omega(1)} \leq \text{dc}_\epsilon(\mathcal{H}) \tag{15}$$

When (15) holds, we can apply Lemma C.8 to conclude that

$$\text{vc}(\mathcal{H})^{\omega(1)} \leq O\left( \frac{s^2 \log^2(|\mathcal{H}| \cdot |X||)}{\epsilon} \right)$$

where $s$ is circuit size of a depth-2 threshold circuit computing Eval$(\mathcal{H})$. Choosing $\mathcal{H}$ appropriately, this implies a super-polynomial lower bound on $s$ for Eval$(\mathcal{H})$.

Thus, it remains to show that for every $\mathcal{H}$, and $\mu$ over $\mathcal{H}$, there exists some constant $\gamma > 0$ such that,

$$\text{vc}(\mathcal{H})^\gamma \leq \text{adc}_{\epsilon,\delta}^b(\mu)$$

To prove this, we negate towards contradiction. Suppose that there exists $\mathcal{H}, \mu$ such that for every constant $\gamma > 0$, it holds that $\text{vc}(\mathcal{H})^\gamma > \text{adc}_{\epsilon,\delta}^b(\mu)$. Then, for $\mathcal{H}$ the class of parities, and $\mu$ the uniform distribution, we get that for every $\gamma$, $\text{adc}_{\epsilon,\delta}^b(\mu) < \text{vc}(\mathcal{H})^\gamma < n^\gamma$. Using this bound on $\text{adc}_{\epsilon,\delta}^b(\mu)$, we can now derive a distributional communication protocol for parities, whose complexity violates known lower bounds of $\Omega(n)$.

The protocol for computing the communication matrix $A(h, x)$ for parities works as follows. The player who owns $x$ samples the embedding $\phi \sim \mathcal{E}$ guaranteed by $\text{adc}_{\epsilon,\delta}^b(\mu) < n^\gamma$. The player then computes $\phi(x)$, and then sends the results to the player who owns $h$. This player then computes the linear combination of $\phi(x)$, which $\epsilon$-approximates $h(x)$, which is also guaranteed by $\text{adc}_{\epsilon,\delta}^b(\mu) < n^\gamma$.

Since $\text{adc}_{\epsilon,\delta}^b(\mu) < n^\gamma$, this protocol for $A$ gives that $A \in \Pi_{c,\gamma}^\zeta$, for $c = \log(b)$ (see section D for definition of $\Pi_{c,\gamma}^\zeta$). Finally, since we stipulated that $b \leq 2^{n^{1-\hat{\epsilon}}}$, we see that $A \in \Pi_{c,\gamma}^\zeta$ for $c = O(n^{1-\hat{\epsilon}})$, which gives the desired contradiction, and concludes the theorem.

$\square$

**Remark.** We note that, by inspection of Adaboost, the weights of the linear combination in our construction in Theorem 4.1 are bounded. Hence, the above theorem indicates that if we want to avoid the complexity-theoretic barrier, then we need a different technique or construction to prove that our construction and relaxation was necessary to get the infinite separation.

## D. Communication Complexity Preliminaries

Let $A$ be a sign matrix (one with entries taking only values in $\{\pm 1\}$). We define the discrepancy of a sign matrix. Let $X, Y$, be two sets indexing the rows and columns of $A$. We write $A(x; y)$ to denote the entry of $A$ indexed by $x \in X, y \in Y$. A *rectangle* $R$ is the set $B \times C$ for $B \subseteq X, C \subseteq Y$. For a fixed distribution $\zeta$ over $X \times Y$, the discrepancy of $A$ with respect to $\zeta$ is defined as

$$\text{disc}_\zeta(A) \triangleq \max_R \left| \sum_{(x,y) \in R} \zeta(x, y) \cdot A(x; y) \right|$$

As a special case, we define discrepancy over *product distributions*

$$\text{disc}^\times(A) \triangleq \min_\zeta \text{disc}_\zeta(A)$$

subject to the constraint that $\zeta$ is product distribution over $X \times Y$.

### D.1. Communication Models

It is well known that discrepancy of a sign matrix is related to the communication complexity of the sign matrix. We define models of communication now.

The 2-party communication model is the following. There are 2 parties, each having unbounded computational power, who try to collectively compute a function. The input to the function is separated into 2 segments, and the $i^{th}$ party sees the $i^{th}$ segment. The parties can send each other direct messages.

Each party may transmit messages according to a fixed protocol. The protocol determines, for every sequence of bits transmitted up to that point (the transcript), whether the protocol is finished (as a function of the transcript), or if, and which, party writes next (as a function of the transcript) and what that party transmits (as a function of the transcript and the input of that party). Finally, the last bit transmitted is the output of the protocol, which is a value in $\{\pm 1\}$. The complexity measure of the protocol is the total number of bits transmitted by the parties.

**Definition D.1** ($\Pi_c$ class). $\Pi_c$ is defined to be the class of functions $f : \{0, 1\} \times \{0, 1\} \to \{\pm 1\}$ that can be computed by a 2-party deterministic communication protocol with complexity $c$.

Frequently it is useful to think of $f$ as being represented by a sign matrix, where $A(x; y) \triangleq f(x, y)$. Hence, we may abuse notation and write that $A \in \Pi_c$.

A model more relaxed than deterministic communication is *distributional* communication.

**Definition D.2** (Distributional $\Pi_c$)**.**  The distributional 2-party communication model allows the protocol to err on certain inputs. Fix a distribution $\rho$ over $\{0, 1\} \times \{0, 1\}$. A function $f : \{0, 1\} \times \{0, 1\} \to \{\pm 1\}$ is in $\Pi^{\zeta}_{c,\gamma}$ if there exists a communication protocol $\pi \in \Pi_c$ such that

$$\mathbb{E}_{(x_1, x_2) \sim \rho} [\pi(x_1, x_2) \cdot f(x_1, x_2)] \geq 2\gamma$$

Distributional communication complexity can be thought of as correlation with $\Pi_c$.

**Definition D.3** (Boolean function correlation)**.**  Define $\text{Cor}(f, \Lambda) \triangleq \max_{h \in \Lambda} |\mathbb{E}[f(x) \cdot h(x)]|$, where $x$ is sampled from a distribution $\zeta$ over the domain.

When we want to measure correlation between two function classes, we have it defined as follows:

**Definition D.4** (Boolean function correlation)**.**  Define $\text{Cor}(\mathcal{C}, \Lambda) \triangleq \min_{f \in \mathcal{C}} \max_{h \in \Lambda} |\mathbb{E}[f(x) \cdot h(x)]|$, where $x$ is sampled from a distribution $\zeta$ over the domain.

### D.2. Discrepancy vs. 2-bit Communication

*Proof of Lemma A.2.*  Let $R'$ be the rectangle that witnesses the maximum value of

$$\left| \sum_{(x,y) \in R'} \zeta'(x, y) \cdot A(x; y) \right|$$

with respect to the worst-case product distribution $\zeta'$. We construct $\pi$ as follows. Alice and Bob, receiving as input $x$ and $y$ respectively, which are sampled according to $\zeta'$, each send a bit indicating whether or not their respective input is contained in $R'$. If both inputs are contained in $R'$, then they output a bit according to the bias over $A$ projected to $R'$ induced by $\zeta'$. Otherwise, they output a random bit.

First, observe that the output of this protocol by definition satisfies

$$\left| \mathbb{E}_{(x,y) \sim \zeta'} [\pi(x, y) A(x; y)] \right| = \sum_{(x,y) \in R} \zeta'(x, y) \cdot \mathbb{E}_{\pi} [\pi(x, y) \cdot A(x; y)]$$

Now, with probability $(p + 1)/2$, for a certain $p \in [-1, 1]$, $\pi(x, y) = A(x; y)$. Thus, for $(x, y) \in R$, $\pi(x, y) \cdot A(x; y) = 1$ with probability $(p+1)/2$. Therefore $\mathbb{E}_{\pi}[\pi(x, y) \cdot A(x; y)] = (p+1)/2 - (1 - (p+1)/2) = (p+1)/2 - 1 + (p+1)/2) = p$. Hence,

$$\left| \mathbb{E}_{(x,y) \sim \zeta'} [\pi(x, y) A(x; y)] \right| = p \sum_{(x,y) \in R} \zeta'(x, y)$$

Note that $p$ is chosen in the proposed protocol specifically so that this implies:

$$\left| \mathbb{E}_{(x,y) \sim \zeta'} [\pi(x, y) \cdot A(x; y)] \right| = \sum_{(x,y) \in R} \zeta'(x, y) A(x; y)$$

Recalling the definition of discrepancy:

$$\text{disc}^{\times}(A) = \left| \sum_{(x,y) \in R'} \zeta'(x, y) \cdot A(x; y) \right|$$

If we apply this definition and the conditions of the theorem, we then get that:

$$\left| \mathop{\mathbb{E}}_{(x,y)\sim\zeta'} \left[ \pi(x,y) A(x;y) \right] \right| \geq \gamma$$

$\square$

### D.3. 2-party Norm

Additionally, an important quantity we will consider is the 2-party norm of a function, $R_2(f)$, which is defined to be the expected product of a function computed on a list of correlated inputs.

**Definition D.5** (2-party norm). For $f : \{0,1\} \times \{0,1\} \to \{\pm 1\}$, the 2-party norm of $f$ is defined as

$$R_2(f) \triangleq \mathop{\mathbb{E}}_{x_1^0, x_2^0, x_1^1, x_2^1 \sim \{0,1\}^n} \left[ \prod_{\epsilon_1, \epsilon_2 \in \{0,1\}} f(x_1^{\epsilon_1}, x_2^{\epsilon_2}) \right] \tag{16}$$

For us, the most important property of $R_2(f)$ is that it upper bounds the correlation of $f$ with functions computable by deterministic 2-party communication protocols. We denote by $\Pi_c$ the set of all $f : \{0,1\} \times \{0,1\} \to \{\pm 1\}$ that have deterministic 2-party communication protocols with cost at most $c$.

Chung & Tetali (1993); Raz (2000); Viola & Wigderson (2007) all demonstrate the following bound:

**Theorem D.6.** *For every function $f : \{0,1\} \times \{0,1\} \to \{\pm 1\}$,*

$$\mathrm{Cor}(f, \Pi_c) = \max_{\pi \in \Pi_c} \left| \mathbb{E}_x \left[ f(x) \cdot \pi(x) \right] \right| \leq 2^c \cdot R_2(f)^{1/4}$$

*for $x$ uniformly distributed over $\{0,1\} \times \{0,1\}$.*

