# OpenReview forum: "The Power of Random Features and the Limits of Distribution-Free Gradient Descent"
_ICML.cc/2025/Conference — ICML 2025 poster_

### Official Review · Reviewer_AcC1 · 2025-03-07

**Overall Recommendation:** 3

**Summary:**

This paper studies the learning capacity of stochastic gradient algorithms. The main results focus on the case of binary classification with either square loss or $01$-loss trained with mini-batch SGD with $c$-approximate clipped gradient. Moreover it is assumed that the data is generated using a function in the considered function class $\mathcal{F}$, ie, we are in a realisable setting. In this setting, it is shown that if the algorithm is able to learn a good predictor regardless of the source data distribution (which is generated by a feature distribution and a function in $\mathcal{F}$), then most of the functions of the function class can be approximated by a linear combination of random features, in a certain sense. This result has two main interpretations: first it means that if we want a gradient algorithm to work on every data distribution, then it not necessary to use models that are more complex than random features. On the other hand, it shows that the function classes that can be learnt from SGD in a distribution-free way mostly contain simple functions. The authors prove their result by extending existing statistical tools, in particular buy introducing a notion of average probabilistic dimension complexity.

**Claims And Evidence:**

The main claims are well explained, first in an informal way and then more formally with proofs.

**Essential References Not Discussed:**

N/A

**Experimental Designs Or Analyses:**

N/A

**Methods And Evaluation Criteria:**

The approach seems to be well-aligned with the existing literature and makes sense for proving the claims.

**Other Comments Or Suggestions:**

N/A

**Other Strengths And Weaknesses:**

Here are a few potential weaknesses:
 - The analysis is limited to binary classification and in particular does not encompass regression problems. The mains theorems are restricted to the square loss, which might be restrictive in the case of binary classification.
 - The setup is restricted to the realisable case, ie, when the function class contains the optimal predictor that has been used to generate the data distribution. This might not be realistic in a modern machine learning setting.

**Questions For Authors:**

- Is it possible to extend the results to cross-entropy loss and / or regression problem?
- Can you explain why SGD with $c$-approximate clipped gradient is considered? Does you results hold with vanilla SGD? This part is not clear to me.
- What is the notation $\sup_{h\leftarrow A}$? The way the error is defined in section 3.2 is not clear to me.
- You mention the $01$-loss, but how can this be used in your results, as you consider gradient-based optimisation?
- Do your results extend to the case where there is label noise, ie, the data is not generated using a deterministic predictor

**Relation To Broader Scientific Literature:**

The authors extend the setting of (Abbe et al., 2021) and the links with the aforementioned paper are made clear.
Moreover, an existing notion of probabilistic dimension complexity from (Kamath et al., 2020) is extended.

**Theoretical Claims:**

I did not check the correctness of the theoretical claims, as the involved tools are a bit far from my expertise.

---

> ### Author Rebuttal · Authors · 2025-03-31
>
> Dear reviewer AcC1,
>
> Thank you for your thoughtful consideration. We will respond directly to the potential weaknesses that you outlined, as well as your questions.
>
> > The analysis is limited to binary classification and in particular does not encompass regression problems. The mains theorems are restricted to the square loss, which might be restrictive in the case of binary classification.
>
> Yes, we consider classification, but our results can extend to other loss functions like bSGD with logistic loss, instead of sq loss. This is true because of the flexibility of the transformation from bSGD to SQ of [1]. Additionally, we do not rule out obtaining an analogous result with sq loss or logistic loss instead of 0-1 loss for the random features. To accomplish that, we could use a different boosting theorem (we use an adaboost theorem). If there is a boosting theorem that we can use to boost weak learners on 0-1 loss to a strong learner under a different loss function, then this should allow one to expand the conclusion to other loss functions.
>
> > The setup is restricted to the realisable case, ie, when the function class contains the optimal predictor that has been used to generate the data distribution. This might not be realistic in a modern machine learning setting.
>
> Our setup is separate from realizable / non-realizable learning. Note, we do not make *any* restrictions on the class $\cal{F}$ of functions (it can be the set of all functions). Instead, complexity is implicitly handled because our theorem’s conclusion (i.e., the size of the random feature representation) is expressed in terms of the number of parameters and gradient steps needed to learn the class $\cal{F}$. For example, if the number of parameters or gradient steps needed to learn $\cal{F}$ is massive due to huge complexity of $\cal{F}$, then the random feature representation would also be massive. A typical example to keep in mind for this case is learning the set of parity functions of the uniform hypercube with bSGD.
>
> **Questions**
>
> > Is it possible to extend the results to cross-entropy loss and / or regression problem?
>
> Yes to cross-entropy, please see the answer to your first weakness above. For regression, it is likely that new techniques would be needed, but we don't think it is inherently impossible.
>
> > Can you explain why SGD with c-approximate clipped gradient is considered? Does you results hold with vanilla SGD? This part is not clear to me.
>
> In short, our theorem is not true for non-clipped gradients. GD with non-clipped gradients is as powerful as general PAC-learning (see [1]), so it’s not possible to convert such algorithms to random features with violating SQ lower bounds.
>
> > What is the notation suph←A? The way the error is defined in section 3.2 is not clear to me.
>
> Supremum is used because we consider worst-case behavior for gradient clipping/rounding. That is, error is with respect to a valid gradient clipping chosen by an adversary at each gradient step. Please see line 195 for explanation of how the gradient clipping works.
>
> > You mention the 01-loss, but how can this be used in your results, as you consider gradient-based optimisation?
>
> The 01 loss is only used to evaluate the random feature representation, which is constructed from the bSGD algorithm. For the bSGD algorithm, indeed the loss must be differentiable such as sq loss or a logistic loss.
>
> > Do your results extend to the case where there is label noise, ie, the data is not generated using a deterministic predictor
>
> We don’t rule out this possibility, but extending it to this setting requires verifying that nothing breaks down across the variety of transformations we have borrowed from other works. For example, the transformation from bSGD to SQ from [1], and SQ lower bounds from [2].
>
> [1] Abbe, E., Kamath, P., Malach, E., Sandon, C., and Srebro, N. On the power of differentiable learning versus pac and sq learning. Advances in Neural Information Processing Systems, 34:24340–24351, 2021.
>
> [2] Blum, A., Furst, M., Jackson, J., Kearns, M., Mansour, Y., and Rudich, S. Weakly learning dnf and characterizing statistical query learning using fourier analysis. In Proceedings of the twenty-sixth annual ACM symposium on Theory of computing, pp. 253–262, 1994.

---

### Official Review · Reviewer_7FE4 · 2025-03-12

**Overall Recommendation:** 3

**Summary:**

This paper proposes a new link between learning a parametric model with mini-batch stochastic gradient descent (SGD) and the approximation of a function class by random features: if a family of functions can be learned, then there exists a distribution of random features (in dimension that is polylogarithmic in all relevant parameters) so that all functions in the family can be well approximated by a linear combination of features.

This is done with a new notion of dimension complexity ("adc"), and through a reduction to statistical query learning.

**Claims And Evidence:**

The authors do their best to show how the theorem is proved, but given that I was not familiar with the papers that the current paper builds on (e.g., the paper by Abbe et al.), it is hard for me to understand the proof and its validity (in a traditional reviewing form, I would put low confidence, but this does not seem to be possible here).

As in most theory papers, the authors also rightfully try to interpret their result. They claim surprise in the proposed equivalence, with the underlying understanding that linear random features are a form a weak learners.

I mostly agree with them with the fact that it is surprising that learnability by mini-batch SGD implies the existence of a random feature distribution. However, the author should insist more on the fact that the distribution of random feature is unknown and depends on the function class in ways unknown to a practitioner. Most of the related work that the paper describes corresponds to explicit random features.

**Essential References Not Discussed:**

N/A

**Experimental Designs Or Analyses:**

N/A

**Methods And Evaluation Criteria:**

N/A (theory paper)

**Other Comments Or Suggestions:**

In the related work section, the description of the work of Chizat et al. , 2019 (on lazy training) is inaccurate: all layers do not move, and not only the bottom layer, showing that the NTK regime is achieved.

**Other Strengths And Weaknesses:**

*

**Questions For Authors:**

(1) Is there any issue with the fact that stochastic gradient descent may have difficulties reaching a global minimum? (I suspect not)
(2) Could you clarify which losses are considered and where, as at the moment it is a bit unclear? (e.g., from line 165, it could be both square and 0/1, but this is probably not the case any more for GD as 0/1 is not differentiable, so the one in line 192 is probably the square loss).
(3) The paper focuses on target function classes with values in {-1,1} and the square loss, which is not the standard in ML. Would this apply to logistic loss as well?

ADDED AFTER REBUTTAL
Thanks for your responses. If the paper ends up being accepted and you have one more page, I would strongly suggest to make it more self contained. It currently requires to know in depth several other papers without much explanation. This would greatly increase its impact.

**Relation To Broader Scientific Literature:**

The paper makes the effort of relating to previous work, but since I don't know that line of work, I can't tell if this is done correctly or not.

**Theoretical Claims:**

Not in details.

---

> ### Author Rebuttal · Authors · 2025-03-31
>
> Dear reviewer 7FE4,
>
> Thank you for your review, we hope to answer your questions below.
>
> > (1) Is there any issue with the fact that stochastic gradient descent may have difficulties reaching a global minimum? (I suspect not)
>
> No, since the statement considers probably approximately correct learnability in terms of the global optimum.
>
> > (2) Could you clarify which losses are considered and where, as at the moment it is a bit unclear? (e.g., from line 165, it could be both square and 0/1, but this is probably not the case any more for GD as 0/1 is not differentiable, so the one in line 192 is probably the square loss).
>
> Right now we are using sq loss for the bSGD learning, since as you say it needs to be differentiable. For the random features, we are using 0-1 loss. However, in both cases, we could potentially consider other losses. To generalize the sq loss in bSGD learning, we can use the flexibility of the conversion from bSGD to SQ of [1]. For flexibility on the 0-1 loss, we could use a different boosting theorem (we use an adaboost theorem). If there is a boosting theorem that we can use to boost weak learners on 0-1 loss to a strong learner under a different loss function, then this should allow one to expand the conclusion to other loss functions.
>
> > (3) The paper focuses on target function classes with values in {-1,1} and the square loss, which is not the standard in ML. Would this apply to logistic loss as well?
>
> Our results start by invoking the transformation of [1], to convert a bSGD algorithm into a related SQ algorithm. In [1], the proof of this transformation considers square loss on labels {0,1}, but the choice of loss function can indeed be modified up to change of technical details in the proof. What is important is the differentiability. So, yes, our results could begin with bSGD on logistic loss instead.
>
> [1] Abbe, E., Kamath, P., Malach, E., Sandon, C., and Srebro, N. On the power of differentiable learning versus pac and sq learning. Advances in Neural Information Processing Systems, 34:24340–24351, 2021.

---

### Official Review · Reviewer_9ucH · 2025-03-12

**Overall Recommendation:** 3

**Summary:**

This paper revisits the power of learning with gradient based methods in a distribution free setting. The main result is to show that if a hypothesis class $F$ can be learnt using gradient descent on some parameterized model with a distribution-free guarantee then for any prior distribution $\mu$ over $F$, there exists a probability distribution over random features such that $h \sim \mu$ with high probability can be approximated by a linear function over those random features; this is formalized in the notion of _average probability dimension complexity_ introduced in the paper.

This is interpreted in the paper by saying that any distribution-free guarantee on learning with gradient descent must be very limited because it implies that _most_ functions in the class can be represented efficiently using a random feature representation.

The proof technique is as follows:
1. A result of Abbe et al. (2021) is invoked to say that learning with gradient descent implies a statistical query learning algorithm, which by standard arguments, implies upper bounds on the _statistical query (SQ) dimension_.
2. The main technical contribution of the paper then is to show an upper bound on average probabilistic dimension complexity in terms of the SQ dimension of the class.

As a corollary, Part 2 resolves a weaker version of a conjecture from Kamath et al. (2021) that asks for an infinite separation between _probabilistic dimension complexity_ and (deterministic) dimension complexity; namely, here an infinite separation is shown between _average probabilistic dimension complexity_ and _dimension complexity_.

The proof for Part 2 involves techniques from communication complexity, a "random feature lemma" and a boosting procedure.

### Post-rebuttal update

Based on the rebuttal discussion, I am increasing my score to 3, since the authors have acknowledged that the concerns raised in this review would be discussed in a revision.

**Claims And Evidence:**

Claims are supported by proofs in the main body or in the Appendix.

**Essential References Not Discussed:**

I think all essential references have been discussed.

**Experimental Designs Or Analyses:**

The paper is entirely theoretical and there are no experiments.

**Methods And Evaluation Criteria:**

The paper is entirely theoretical, so I believe this question is not relevant.

**Other Comments Or Suggestions:**

I think the notation of average probabilistic dimension complexity would simpler with a single parameter $\epsilon$, defined as, $\\mathrm{adc}\_\\epsilon(\\mu)$ equal to the smallest $d$ such that there exists a distribution $\\mathcal{E}$ over $d$-dimensional representations such that $\\mathbb{E}\_{h \\sim \\mu} \\sup\_{\\rho \\in \\Delta(X)} \\mathbb{E}\_{\\phi \\sim {\\cal E}} \\inf\_{w} {\\cal L}^{\\mathcal{D}\_{h, \\rho}}\_{01}(\\langle w, \\phi \\rangle) \le \epsilon$. It is possible to convert to the two parameter version by using Markov's inequality, since only the case of small constant $\delta$ is considered.

Incidentally, there is [another paper](https://arxiv.org/abs/2411.10784) that also tries to address the conjecture in Kamath et al. (2021), but it does it in a very different way, by considering partial functions. I believe the techniques are also very different. But it might still be interesting to discuss this paper in the context of the conjecture.

I was looking at the paper of Kamath et al. (2021), and they have a notion of probabilistic distributional dimension complexity $\\mathrm{dc}\_{\\epsilon}^{\\mathcal{D}}$ and they show an "infinite gap" between this and deterministic dimension complexity. Is there any relation between $\\sup\_{\\mathcal{D}} \\mathrm{dc}\_{\\epsilon}^{\\mathcal{D}}(F)$ and $\\sup_{\\mu} \\mathrm{adc}_{\\varepsilon}(F)$ (as in the one-parameter version of $\\mathrm{adc}$ defined above)? Maybe there isn't, but I was just wondering.

### Minor comments
* Line 215 (left): There seems to be some typo in definition of $\phi_t : X\\{\pm 1\\} \to [-1, 1]$

**Other Strengths And Weaknesses:**

### Strengths
I think the paper shows a nice result that average probabilistic dimension complexity can be upper bounded in terms of statistical query dimension. While the techniques are borrowed from powerful techniques from prior work, I feel there is some novelty in this technical contribution.

### Weaknesses

I already raised a concern about Theorem 3.3 (main theorem) regarding the condition $bc^2 \\ge \\Omega(\\log Tp/\\delta)$ (see "Theoretical Claims" section). I feel this breaks the narrative about "gradient descent" in the paper.

But beyond that, I feel the notion of average probabilistic dimension complexity is not _that_ well motivated. In particular, it does not really help with learning a worst case hypothesis in the class, which is the usual setting of learning. All this notion says is that for any prior $\mu$ over the hypothesis class, there exists a random features representation such that most functions in the class are well approximated. But since this random features representation depends on $\mu$, this does not lead to learning algorithm. So, from that point of view, I don't fully buy the narrative that having small average probabilistic dimension complexity suggests any limitation of the hypothesis class, even though, I agree that parities are hard even in this sense.

**Questions For Authors:**

I would like to hear from the authors regarding the weaknesses pointed out above.

**Relation To Broader Scientific Literature:**

The paper introduces a novel notion of average probabilistic dimension complexity, and shows that it can be upper bounded in terms of the SQ dimension. This is the main result in the paper.

**Theoretical Claims:**

Proofs have been provided for all theoretical claims in the paper.

One thing that I feel is problematic is: In Theorem 3.3, why is the condition $bc^2 \\ge \\Omega(\\log Tp/\\delta)$ not required? The proof seems to simply invoke Theorem 3.4 as the first step, which does seem to have this condition. If this condition is indeed required, then I feel the narrative of the paper changes quite a bit. The story is not really about _any_ gradient descent based method, but only one with "low precision" (as defined in Abbe et al. (2021)). In that sense, the paper is really about statistical query learning methods, and the "gradient descent" part is not providing that much insight.

---

> ### Author Rebuttal · Authors · 2025-03-31
>
> Dear reviewer 9ucH,
>
> We appreciate your thoughtful review. We will respond to specific weaknesses that you addressed in your review.
>
> > One thing that I feel is problematic is: In Theorem 3.3, why is the condition bc^2≥Ω(log⁡Tp/δ)
>  not required?
>
> Yes, it is required. This restriction is introduced in the text preceding the theorem (top of paragraph 2 in section 3.2 - line 238). We will include this inside the theorem for extra clarity.
>
> > If this condition is indeed required, then I feel the narrative of the paper changes quite a bit. The story is not really about any gradient descent based method, but only one with "low precision" (as defined in  [4]). In that sense, the paper is really about statistical query learning methods, and the "gradient descent" part is not providing that much insight.
>
> We believe that your interpretation is technically correct in some sense. However, there is a large body of work that studies limitations of gradient descent through SQ lower bounds. For a quintessential example, see [5].
>
> Additionally, we want to point out that expecting this type of result to apply to any gradient descent based method is not well-founded. This is because our theorem *does not hold* when there is no restriction on the precision of the gradients. To see this, we can borrow from [4], who show in theorem 1a of their paper that (distribution-free) PAC-learning algorithms can be simulated by (distribution-free) bSGD algorithms, when allowed fine enough gradient precision c (depending on the batch size b, specifically, c < 1/8b; note, in their paper they use \rho to denote gradient precision). Using this theorem, it follows that for (b,c) such that c < 1/8b, then bSGD can learn parities in the distribution-free case. From here, we can conclude that our transformation from bSGD to random features cannot hold for (b,c) such that c < 1/8b, since the random feature representation would violate SQdim lower bounds for parities.
>
> We do think this discussion is insightful and plan to discuss this in the final version of this paper.
>
> > I feel the notion of average probabilistic dimension complexity is not that well motivated... All this notion says is that for any prior μ  over the hypothesis class, there exists a random features representation such that most functions in the class are well approximated. But since this random features representation depends on μ, this does not lead to a learning algorithm. So, from that point of view, I don't fully buy the narrative that having small average probabilistic dimension complexity suggests any limitation of the hypothesis class
>
> It is true that the order of quantifiers makes it so that “the learner must know the prior μ” so-to-speak. This is why we interpret the theorem, as explained in the introduction, as support for the the statement:
>
> *If a function class is learnable in a distribution-free manner by gradient descent (with some restricted precision), then most functions in the class must have a relatively simple random feature representation.*
>
> This statement is fully supported by our theorem if you fix μ to be the uniform distribution, as this means most functions in the class (i.e., a large fraction of the class, or any large subset of the class) have a small random feature representation. Moreover, **we must emphasize** that the fact that any large subset of the class has a random feature representation is a major limitation. Many works give relevant examples of classes that **cannot** be represented in such a way at all: [1] [2] [3]. [1] does this for parity functions on k bits, [2] does (single) ReLUs, and [3] does “ResNets.” **Hence, the fact that we show such a scheme *exists* at all is a strong statement.**
>
> On the other hand, as you pointed out, when μ is “unknown,” there is no explicit feature distribution. Thus, we do not interpret the theorem as saying that  gradient descent can be replaced by random feature learning algorithms. This is touched on in the first page of the introduction, but we are happy to expand this discussion in the final version of the paper for maximum clarity.
>
> References
>
> [1] Amit Daniely and Eran Malach. Learning parities with neural networks. arXiv preprint arXiv:2002.07400, 2020.
>
> [2] Gilad Yehudai and Ohad Shamir. On the power and limitations of random features for understanding neural networks. In Advances in Neural Information Processing Systems, pages 6598–6608, 2019.
>
> [3] Zeyuan Allen-Zhu and Yuanzhi Li. What can resnet learn efficiently, going beyond kernels? In Advances in Neural Information Processing Systems, pages 9017–9028, 2019.
>
> [4] Abbe, E., Kamath, P., Malach, E., Sandon, C., and Srebro, N. On the power of differentiable learning versus pac and sq learning. Advances in Neural Information Processing Systems, 34:24340–24351, 2021.
>
> [5] Surbhi Goel, Gollakota, A., Jin, Z., Karmalkar, S., Klivans, A. Superpolynomial Lower Bounds for Learning One-Layer Neural Networks using Gradient Descent. arXiv:2006.12011, 2020.

---

### Official Review · Reviewer_NhLC · 2025-03-19

**Overall Recommendation:** 3

**Summary:**

This paper investigates how powerful are gradient-based algorithms on some differentiable models for **distribution-free** learning settings. The main theoretical result (see Theorem 3.2) is: essentially, whenever the function class is SQ learnable or differentiably learnable in the sense of [Abbe et al. 2021] then it is also learnable with some random feature kernel up to polynomial blow-ups. So the result essentially establishes that without making any assumptions on the input distribution, it is not possible to show exponential separations between Gradient algorithms and random feature kernels.

The only small caveat, and perhaps necessary to show this result, is that the guarantee holds with high probability over the function class for an arbirary prior. But does not hold for any function in the class. Perhaps, this is inherently necessary to show the result.

So to show such guarantees, the paper introduces another notion of average complexity measure--average probabilistic dimension complexity (adc)---which is shown to be polynomially related to statistical query dimension, to achieve the desired claim.

**Claims And Evidence:**

Yes.

**Essential References Not Discussed:**

NA

**Experimental Designs Or Analyses:**

NA

**Methods And Evaluation Criteria:**

NA

**Other Comments Or Suggestions:**

Please see the weaknesses. Also, though I don't have line-by-line suggestions, see if the introduction can be made more effective overall.

Can you please make the dependence on \ell explicit throughout in the notation of adc itself, like Kamath et al? Though it has been specified several times clearly, I suggest you create a notation where this is explicit.

Also, the square loss notation can be ell_{sqr} rather than capital SQ, which is relevant for the statistical query. Please try to make this different if possible.

IMPORTANT: doesn't the centered statement in lines 52-53, right side at the end of page 1 summarizing the result, create a wrong impression of the order of quantifiers? This is also used in TL;DR. Perhaps, it should have been:
"...., then there exists a random feature (depending on a prior) that can express most functions under the prior." or make it even more clear:
"...., then for any prior over function class, there is a random feature model that can express most functions under that prior."

**Other Strengths And Weaknesses:**

See summary for strengths. The paper has a good theoretical contribution.

Weaknesses: Overall, the paper's exposition can be improved.
Line 21: "though the distribution free learning is the desired goal." I strongly disagree with this statement. I don't think this is the ultimate goal. The goal is to come up with the right assumptions that can inform us why learning works. And it is well-known that distribution-free settings are too pessimistic due to the worst-case hardness results. This is never taken as the ultimate goal.
"The "theoretical value of this result" paragraph can be improved.
1. Line 64-67 "a common finding...learning outcomes": The statement does not read well. It is not that assumption enables better learning outcomes. Perhaps a better way to put this point forward, which I did not find very clearly in the entire paper either, is as follows. "Input data comes from a specific distribution. However, the distribution-independent settings are easier to discuss and have been extensively studied. However, indeed, the lower bounds on this should not be seen as the fundamental barrier, as this bounds are too pessmistic, i.e. it holds over some worst-case distribution. The result in this paper makes this formal in the context of kernel vs gradient-based method on NNs." This can be also said in the "This work in a nutshell" paragraph before the last line.
2. The entire 2nd (line 76-86) on general references on parity can be either completely removed (or delayed if authors mention these works anyway). But they don't add anything to the point. It hurts the flow of reading to the most important paragraph to follow after that. Also, in lines 100-101, in what distribution-specific setting parties are not hard to learn? Generally, uniform distribution is also seen as the distribution-specific setting, so it is better to specify the distribution as well.

**Questions For Authors:**

1. Is the 0-1 loss in line 254 defined incorrectly? Should it have \neq instead of equal-to? This mistake is also there at other places.
2. Can you please walk me through the argument as to why this prior mu over function is necessary for this type of result to hold? What is the main intuition on why this could be necessary?
3. Why do you ell_sq for bSGD, but the result for kernel is shown over 0-1 loss? I understand that for bSGD, you need the loss to be an almost differentiable one, so can you show this for general surrogate losses? More importantly, can you show the result for random feature approximation for, let's say sqr loss? Can you explain to me a bit about this discrepancy and to what extent it could be avoided?

**Relation To Broader Scientific Literature:**

When NN trained with gradient-based algorithms are more powerful than kernel methods is an important question in learning theory.  At least hundreds of works, if not more, have looked at this question from several different points of view. The results of this paper are essentially saying that the assumption on the input distributions is necessary to understand this separation. Otherwise, if the goal is to succeed over all distributions, then this separation collapses, and they are the same up to polynomial factors. I think the findings are new.

**Theoretical Claims:**

I havn't checked. But the claims seem to be correct and enough discussed in the main text.

---

> ### Author Rebuttal · Authors · 2025-03-31
>
> Dear reviewer NhLC,
>
> Thank you for your thoughtful review. We will directly respond to the weaknesses and questions that you outlined.
>
> > Overall, the paper's exposition can be improved. Line 21: "though the distribution free learning is the desired goal." I strongly disagree with this statement. I don't think this is the ultimate goal. The goal is to come up with the right assumptions that can inform us why learning works. And it is well-known that distribution-free settings are too pessimistic due to the worst-case hardness results. This is never taken as the ultimate goal.
>
> Thank you for this feedback. Line 21 says “While distribution-free learning is a desirable goal, it is often computationally challenging, as it requires the algorithm to handle worst-case scenarios.” We feel that this line is meant to say something different than the way it was interpreted, so we will modify it to prevent this. We do not intend to weigh in on what the “ultimate goal” is (or should be), and we just meant to say that distribution-free algorithms can be desirable (in a literal sense), and add the context that it is often computationally challenging to achieve.
>
> > Line 64-67 "a common finding...learning outcomes": The statement does not read well...
>
> Thank you for pointing it out, and we intend to improve it. To improve it, we like your suggested point as a template, and will include something similar in the “this work, in a nutshell” paragraph. E.g.,
>
> Input data comes from a specific distribution. However, the distribution-independent settings are easier to discuss and have been extensively studied.Yet, hardness results in this setting should not be seen as a fundamental barrier. Taken at face-value, they only apply to some worst-case distribution. In this paper, we make this formal in the context of kernel vs gradient-based method on NNs, by showing that ...
>
> > The entire 2nd (line 76-86) on general references on parity can be either completely removed...
>
> We will move this information elsewhere. Indeed, it may break the flow, but for some readers less familiar with parities, and the study of parities, it may be helpful context.
>
> > in what distribution-specific setting parties are not hard to learn? Generally, uniform distribution is also seen as the distribution-specific setting, so it is better to specify the distribution as well.
>
> [1] Consider a family distributions where parities are not hard. As a special case, they consider a biased product distribution over the hypercube. We will specify this as an example.
>
> > IMPORTANT: doesn't the centered statement in lines 52-53, right side at the end of page 1 summarizing the result, create a wrong impression of the order of quantifiers?
>
> Using the phrasing then **most** functions in the class must have a relatively simple random feature representation alludes to the case where the prior μ is fixed to uniform. In this case there is one random feature distribution constructed by our proof. So while our statement is correct, it may not be clear, and we definitely do not want to create the wrong impression. We will clarify in the final version.
>
> Questions:
>
> 1. Yes (latex rendering error)
>
> 2. The main intuition is that in the proof of the random feature lemma (page 10), we use the ability to sample new concepts from the distribution. This seems unavoidable to apply our communication complexity technique which operates in the “distributional” communication model. While our techniques basically operate only in this setting, it is not clear if it is necessary.
>
> 3. Yes, it can hold for more general loss functions than sqr loss, using the flexibility of the theorem of [2], which converts bSGD to SQ. For the random feature representation, our communication complexity technique produces random features that are essentially weak learners for the target distribution w.r.t. 0-1 loss. However, if there is a boosting theorem that we can use to boost such weak learners to a strong learner under a different loss function, then this should allow one to expand the conclusion on the random feature representation to other loss functions.
>
> [1] Malach, E. and Shalev-Shwartz, S. Learning boolean circuits with neural networks. arXiv preprint arXiv:1910.11923, 2019.
>
> [2] Abbe, E., Kamath, P., Malach, E., Sandon, C., and Srebro, N. On the power of differentiable learning versus pac and sq learning. Advances in Neural Information Processing Systems, 34:24340–24351, 2021.

---

> > ### Comment · Reviewer_NhLC · 2025-04-08
> >
> > So do authors believe this limitation of prior mu is necessary for showing such distribution-free equivalence between SQ methods and random feature methods? Or is it just the limitation coming from the current techniques used? I could not understand clearly from the response.
> >
> > So far, I was under the impression that this seems to be fundamentally necessary due to lines 142-146. How I should interpret these lines?

---

> > > ### Author Response · Authors · 2025-04-08
> > >
> > > Lines 142-146 are:
> > > > Our relaxed notion of average probabilistic dimension complexity is sufficient for an affirmative resolution, but we also show that there may exist complexity theoretic barriers to demonstrating that our relaxed notion is necessary for the separation.
> > >
> > > "sufficient for an affirmative resolution" refers to Corollary C.3, which is the result showing that there is $\cal{H}, \mu$ such that $adc(\mu)$ is independent of $n$, while $dc(\cal{H})$ is exponential in $n$:
> > >
> > > Corollary C.3
> > > > There exists a hypothesis class $\cal{H}$, with domain $\{\pm 1\}^n$ and range $\{\pm 1\}$, which satisfies for 0/1 loss and \textbf{any} prior distribution $\mu$ over $\cal{H}$, and arbitrarily small constant $\delta > 0$:
> > >
> > > - $dc(\cal{H}) \in 2^{\Omega(n^{\frac{1}{4}})}$
> > > - $adc(\cal{H}) \in O(1/\epsilon)$.
> > >
> > > Perhaps it is useful to now restate lines 142-146 for clarity:
> > > > Our relaxed notion of average probabilistic dimension complexity is sufficient to prove corollary C.3. It would be nice to show that it is also *necessary* to prove C.3. But, in theorem C.7, we show that under certain restrictions on the weights of the random features$^1$, proving that the relaxed notion is *necessary* for this separation implies an explicit super-polynomial depth-2 threshold circuit lower bound, which resolves a major open problem in circuit complexity theory.
> > >
> > > Here, "*necessary* to prove C.3" means showing there exists $\cal{H}, \mu$ such that $adc(\mu)^{\omega(1)} < dc _\epsilon(\cal{H})$. In theorem C.7, we show that proving this inequality resolves the circuit lower bound question.
> > >
> > > **conclusion/tldr:** All in all, we don't know if the prior $\mu$ is necessary. We do show that, when using the current techniques (broadly construed), then *proving* the prior $\mu$ is necessary is very hard -- since it resolves a major open conjecture in complexity theory.
> > >
> > > For what it's worth, *proving* the prior $\mu$ is necessary would resolve the complexity conjecture in the "expected" way, so our theorem C.7 **does not** indicate that the prior $\mu$ is not necessary -- only that *proving* that it is necessary is difficult. We view lines 142 -146, and the accompanying theorem C.7, as fairly supplementary to the main results and body of work in this paper.
> > >
> > > Footnotes
> > > 1. Our construction satisfies these restrictions.

---

### Decision · Program_Chairs · 2025-05-01

**Decision:**

Accept (poster)

**Comment:**

This paper proves a novel theoretical connection between a model’s optimizability by gradient descent and its approximability by random features, characterized by a new average variant of the probabilistic dimension complexity and using tools from statistical-query learning and community complexity theories. In a nutshell, it shows if a differentiable parametric model can be successfully optimized by a distribution-free and unnecessarily very precise mini-batch gradient descent technique, then the model can be well approximated by a linear combination of random features with high probability (w.r.t. a prior distribution on the model space) where the combination size is upper bounded by a polynomial function of the descent step size. This can be interpreted as a fundamental limitation of distribution-free learning i.e., if a model class is learnable by distribution-free gradient descent, then it is approximable by a random feature representation and thus inherits its known limited expressive power.

---

All reviewers find the paper acceptable, and most commend the theoretical contribution. A notable issue raised by reviewers NhLC/7FE4/AcC1 is the choice of squared loss instead of 0-1 loss for measuring classification error. In rebuttal, authors explain this is due to the invoking of a prior result that transforms gradient descent to statistical-query learning which considers squared loss, and clarify more general loss functions can be chosen as long as they are differentiable. In addition, reviewer 9ucH suggests the condition of `low precision’ is not sufficiently discussed which may change the narrative quite a bit, but raises his score after authors provide more explanations in rebuttal and promise to add more in the final paper.

Overall, the paper provides an interesting and novel theoretical insight backed up by solid and novel theoretical justifications. I think its contribution outweighs the slightly restrictive (yet seemingly extendible) setting and a few clarity issues in presentation. The paper could be accepted if there is a room in the program. Should it be accepted, authors are encouraged to incorporate all feedback in revision, especially regarding the discusssions on 0-1 loss and low-precision condition -- perhaps a more appropriate claim is a model optimizable by low-precision gradient descent has limited expressive power?

---

p.s. While the work is well contextualized in the deep learning literature, I’m curious if it may have any connection to the broader random feature literature e.g., [1,2]. (not a must)

[1] Weighted Sums of Random Kitchen Sinks: Replacing minimization with randomization in learning, NeurIPS 2008.

[2] Generalization Properties of Learning with Random Features, NeurIPS 2017.